# Stress resets ancestral heritable small RNA responses

Leah Houri-Zeevi†*, Guy Teichman†*, Hila Gingold, Oded Rechavi*

Department of Neurobiology, Wise Faculty of Life Sciences & Sagol School of Neuroscience, Tel Aviv University, Tel Aviv, Israel

**Abstract** Transgenerational inheritance of small RNAs challenges basic concepts of heredity. In *Caenorhabditis elegans* nematodes, small RNAs are transmitted across generations to establish a transgenerational memory trace of ancestral environments and distinguish self-genes from non-self-elements. Carryover of aberrant heritable small RNA responses was shown to be maladaptive and to lead to sterility. Here, we show that various types of stress (starvation, high temperatures, and high osmolarity) induce resetting of ancestral small RNA responses and a genome-wide reduction in heritable small RNA levels. We found that mutants that are defective in various stress pathways exhibit irregular RNAi inheritance dynamics even in the absence of stress. Moreover, we discovered that resetting of ancestral RNAi responses is specifically orchestrated by factors that function in the p38 MAPK pathway and the transcription factor SKN-1/Nrf2. Stress-dependent termination of small RNA inheritance could protect from run-on of environment-irrelevant heritable gene regulation.

*For correspondence:
leah.houri@gmail.com (LH-Z);
guy.teichman@gmail.com (GT);
odedrechavi@gmail.com (OR)

†These authors contributed equally to this work

**Competing interests:** The authors declare that no competing interests exist.

## Introduction

Different human diseases, such as several imprinting-associated syndromes (Angelman syndrome, Prader-Willi syndrome, and Beckwith-Wiedemann syndrome), arise due to the inheritance of parental information that is not encoded in the DNA sequence (*Tucci et al., 2019*). Furthermore, and although the underlying mechanisms are still unclear, many widespread disorders were suggested to be influenced by non-genetic inheritance and to be affected by the ancestors' life history (*Bohacek and Mansuy, 2015*; *Chen et al., 2016*; *Gapp et al., 2014*; *Kazachenka et al., 2018*; *Nilsson et al., 2018*; *Öst et al., 2014*; *Skvortsova et al., 2018*; *Teperino et al., 2013*). Development of methods for resetting heritable effects could potentially benefit the future treatment of these diseases and enable the progeny to start as a 'blank slate'.

In *Caenorhabditis elegans* nematodes, much knowledge has been gained regarding the mechanisms that enable transgenerational gene regulation via inheritance of small RNAs (*Lev and Rechavi, 2020*; *Rechavi and Lev, 2017*; *Serobyan and Sommer, 2017*). Small RNA inheritance factors, which are specifically required for the regulation of heritable responses, have been identified (*Ashe et al., 2012*; *Buckley et al., 2012*; *de Albuquerque et al., 2015*; *Houri-Ze'evi et al., 2016*; *Lev et al., 2019b*; *Lev et al., 2017*; *Shirayama et al., 2012*; *Wan et al., 2018*; *Xu et al., 2018*), and specific worm Argonaute proteins (HRDE-1, CSR-1, WAGO-4, WAGO-1) were found to physically carry small RNAs in the germline and across generations (*Ashe et al., 2012*; *Claycomb et al., 2009*; *Shirayama et al., 2012*; *Wedeles et al., 2013*; *Xu et al., 2018*). The multigenerational small RNA inheritance response requires RNA-dependent RNA Polymerases (RdRPs) which use the target mRNA as a template for amplifying 'secondary' or 'amplified' small RNAs (*Rechavi et al., 2011*; *Sapetschnig et al., 2015*). The amplification reaction outcompetes the dilution of the heritable small RNA molecules in every generation and enables the transgenerational transmission of small RNAs (*Rechavi et al., 2011*). Amplification of heritable small RNAs can be induced by multiple 'primary' small RNA species, of both exogenous and endogenous sources – such as small interfering RNAs and PIWI-interacting RNAs (*Das et al., 2008*; *Sijen et al., 2001*). Amplification of transgenerationally

inherited small RNAs occurs in germ granules (*Dodson and Kennedy, 2019*; *Lev et al., 2019b*; *Ouyang et al., 2019*), cytoplasmic condensates made of RNA and proteins found in germ cells of many organisms (*Brangwynne et al., 2009*; *Shin and Brangwynne, 2017*; *Voronina et al., 2011*). In the nucleus, amplified small RNAs lead to transcriptional silencing of their targets in cooperation with chromatin regulators (*Burton et al., 2011*; *Kalinava et al., 2017*; *Lev et al., 2019a*; *Lev et al., 2017*; *She et al., 2009*). Nuclear small RNAs promote modification of chromatin, and some changes in histone marks are transgenerationally inherited, also in response to environmental changes (*Klosin et al., 2017*).

The worm's small RNA pools can change transgenerationally in response to multiple environmental challenges such as viral and bacterial infection (*Kaletsky et al., 2020*; *Moore et al., 2019*; *Rechavi et al., 2011*), starvation (*Ewe et al., 2020*; *Rechavi et al., 2014*), and stressful temperatures (*Ni et al., 2016*; *Schott et al., 2015*). Further, *C. elegans* actively regulates small RNA inheritance and controls the duration and potency of the transgenerational effects across generations (*Houri-Ze'evi et al., 2016*; *Houri-Zeevi and Rechavi, 2017*). Heritable RNA interference (RNAi) responses, which are mediated by small RNAs, can be induced by targeting germline-expressed genes using double-stranded RNA (dsRNA) triggers. Typically, at the population level, such heritable responses last three to five generations (*Alcazar et al., 2008*), but the duration of the heritable response varies among different individuals (*Houri-Zeevi et al., 2020*) and in mutants of epigenetic factors (*Houri-Ze'evi et al., 2016*; *Perales et al., 2018*; *Spracklin et al., 2017*). For example, MET-2, a putative histone 3 lysine 9 (H3K9) methyltransferase and the homologue of mammalian SETDB1, is required for termination of heritable RNAi responses and reestablishment of the zygote's epigenetic ground state. Accordingly, RNAi inheritance is stable in *met-2* mutants and is not diminished across generations (*Kerr et al., 2014*; *Lev et al., 2017*). Even in wild-type animals, the inheritance of ancestral RNAi responses can be extended by triggering dsRNA-induced silencing of other genes in the progeny (*Houri-Ze'evi et al., 2016*) and by selecting lineages of worms which have stronger heritable effects (*Houri-Zeevi et al., 2020*). Together, these different mechanisms constitute a transgenerational 'timer' that restricts the inheritance of small RNA responses across generations (*Houri-Zeevi and Rechavi, 2017*).

It was hypothesized that the continuation of some gene expression programs in the progeny could increase the descendants' chances to survive, especially if parents and progeny experience the same conditions (*Houri-Ze'evi et al., 2016*; *Jablonka, 2017*; *Jablonka, 2013*; *Kishimoto et al., 2017*). However, if environmental conditions change, the carryover of ancestral responses could become detrimental (*Jablonka, 2013*). Indeed, in worms, mutants that are unable to regulate the multigenerational accumulation of heritable small RNAs become sterile (*Lev et al., 2019b*; *Lev et al., 2017*; *Ni et al., 2016*; *Simon et al., 2014*). It is therefore possible that mechanisms have evolved to terminate or extend small RNA-based inheritance according to the presence or absence of dramatic shifts in settings.

In this study, we examined if and how changes in growth conditions between generations alter the dynamics of parental heritable small RNA responses. We found that stress, and not any change in growth conditions across generations, resets small RNA inheritance and decreases the general pools of small RNA in the worms. The termination of inheritance following stress is orchestrated specifically by the p38 MAPK stress pathway and the SKN-1/Nrf2 transcription factor. Interestingly, resetting is canceled in mutants of the *met-2*/SETDB1 putative H3K9 methyltransferase. We suggest that the mechanisms of stress-induced resetting of ancestral responses might enable the worms to better cope with newly introduced environmental challenges.

## Results

We tested how a mismatch in the growth conditions of parents and progeny affects small RNA inheritance across generations. To this end, we examined how exposure to stress at the next generations after the initiation of a heritable response, affects the course of inheritance. To monitor small RNA-mediated inheritance, we used three different inheritance assays: exogenous dsRNA-derived inheritance, endo-siRNAs-derived inheritance, and piRNA-derived inheritance.

## Stress resets exogenous dsRNA-derived heritable silencing in a transgenerational manner

First, we investigated the effect of stress on heritable responses initiated by exogenously derived small RNAs. We used worms that carry an integrated single-copy *gfp* transgene, under the control of the P*mex-5* promoter (germline expression, see Materials and methods). Feeding the worms with bacteria that express anti-*gfp* dsRNA induces heritable silencing of the *gfp* transgene for ±3–5 consecutive generations (*Houri-Ze'evi et al., 2016*) (see Materials and methods). We initiated a heritable anti-*gfp* silencing response at the parental generation (P0) and then exposed the next generation (F1) to three different types of stress. After examining several different stress regimes (namely different magnitudes and durations of stress, see Materials and methods), we chose to either heat shock the worms for 2 hours (hereon, Heat stress) (*Zevian and Yanowitz, 2014*), culture them in hyperosmotic conditions for 48 hours (hereon Osmotic stress) (*Rodriguez et al., 2013*), or starve them for 6 days (hereon Starvation stress) (*Rechavi et al., 2014*) (See *Figure 1A and B*, *Figure 1—figure supplement 1*, and Materials and methods).

We scored the potency of the heritable silencing response both at the stressed generation (F1) and at the following generations which did not experience stress (F2-F6). We found that stress during the L1 stage led to a strong reduction of heritable *gfp* silencing within the same generation (F1) and in the next generations that were not directly exposed to stress (F2-F5, in the F6 generation the heritable response was generally lost in all groups – in accordance with the bottleneck of inheritance. See *Figure 1*, and *Figure 1—figure supplement 3*). We observed similar results using a different single-copy *gfp* transgene under the P*pie-1* promoter (see *Figure 5—figure supplement 2C*).

We then tested whether the ability of stress to reset heritable small RNA responses depends on the developmental stage or the generation in which stress is applied: We found that, as was observed when stress was applied during the L1 stage, stress during adulthood leads to resetting of heritable silencing in the next generation (q < 0.0001, *Figure 1—figure supplement 4*). In contrast, when stress was applied two generations after the parental exposure to dsRNA (P0: initiation of inheritance, F1: no stress, F2: stress), we did not detect statistically significant stress-induced resetting (*Figure 1—figure supplement 5*). We previously showed that the fate of the heritable response (its persistence across generations) is determined at the first generations (*Houri-Zeevi et al., 2020*) and that re-challenging the F1 generation – but not the F2 generation – with RNAi extends the duration of ancestral RNAi responses (*Houri-Zeevi and Rechavi, 2017*). Together, these results indicate the existence of a critical period at the F1 generation, during which the heritable response is still plastic and can be modified by external perturbations.

## Stress resets endogenously derived heritable small RNA silencing within the same generation

Next, in the second and third sets of assays of small RNA inheritance, we examined if stress can also reset heritable silencing that is triggered by *endogenous* small interfering RNAs (endo-siRNAs) or PIWI-interacting small RNAs (piRNAs). For this purpose, we used (1) worms that carry a *gfp* transgene that contains an endo-siRNA-target sequence (a genomically integrated 'endo-siRNA' sensor *Billi et al., 2012*), and (2) worms that stochastically silence a foreign *mcherry* transgene, that contains multiple piRNAs-recognition sites (*Zhang et al., 2018*) (see also *Figure 2A and B*, and Materials and methods). We found that all three stressors (Heat, Osmotic, and Starvation stress) reset both endo-siRNAs- and piRNAs-mediated heritable silencing in worms that were directly exposed to stress (Dunn's test, q-value <0.0005 and q-value <0.0007, respectively. See *Figure 2A and B*). However, in contrast to exo-siRNAs-mediated silencing that was reset in a transgenerational manner (*Figure 1C*), we found that unstressed worms in the next generations re-established endo-siRNAs- and piRNAs-mediated silencing. Unlike exogenous primary small RNAs which cannot be re-synthesized in the progeny, primary endo-siRNAs and piRNAs are encoded in the genome and do not depend on exogenous sources for their existence (*Ambros et al., 2003*; *Cecere et al., 2012*; *Duchaine et al., 2006*; *Gu et al., 2012*; *Lee et al., 2006*; *Lemmens and Tijsterman, 2011*; *Ruby et al., 2006*). The re-establishment of endo-siRNAs and piRNAs-mediated silencing in the next generations after stress indicates that these small RNAs can be transcribed de novo at each generation, and thus can compensate for stress-induced erasure of parental small RNA molecules,

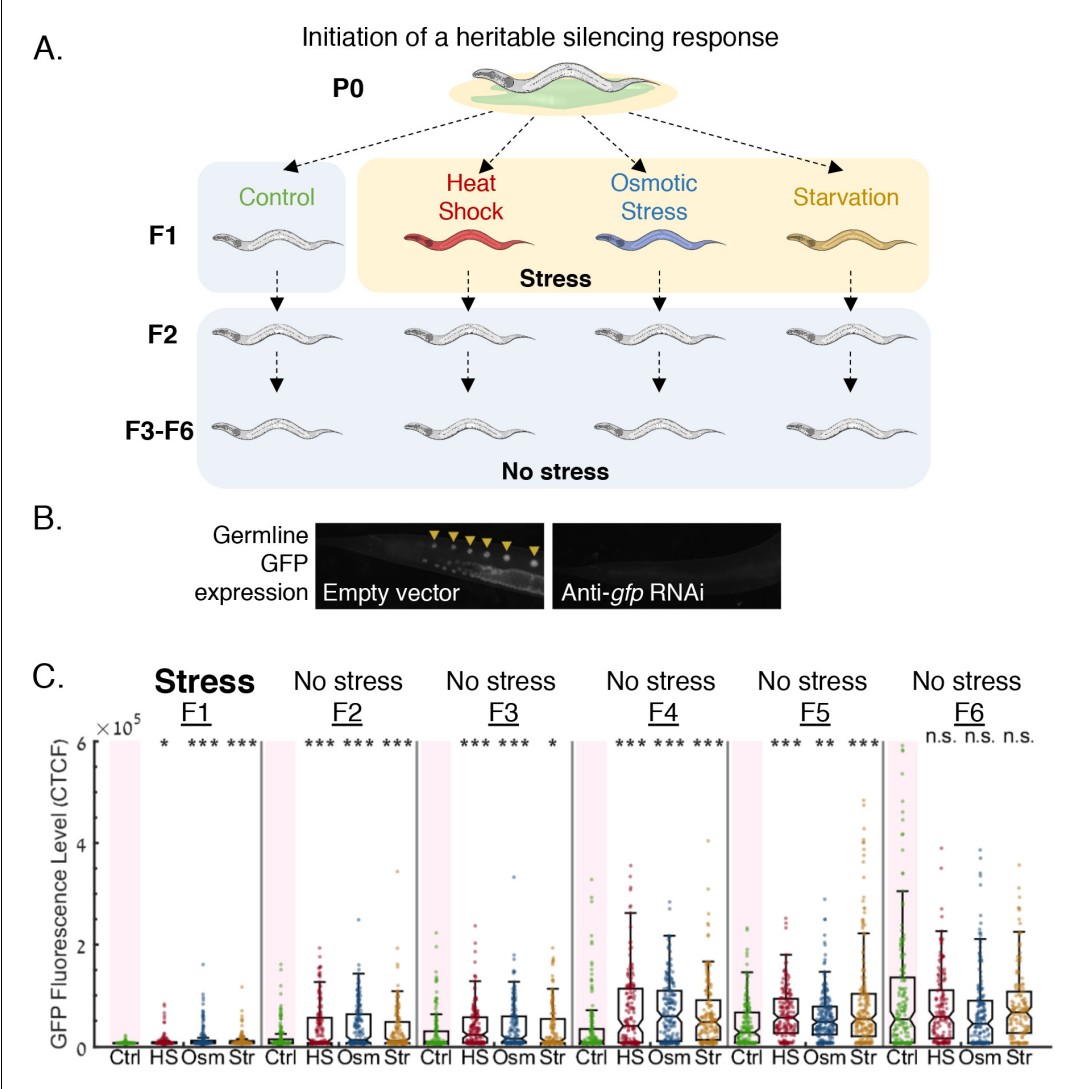

**Figure 1.** Stress resets heritable small RNA silencing. (**A**) *Experimental scheme*. Heritable small RNA responses are initiated at the first generation, and the F1 progeny are then subjected to three different stress types (heat shock {HS}, hyperosmotic stress {Osm}, starvation {Str}). Inheritance of the ancestral response is scored both in the stressed generation and in the next generations which were grown under regular growth conditions. (**B**) Representative images of worms containing the *Pmex-5::gfp* transgene, treated with empty vector containing bacteria (left) or with anti-*gfp* dsRNA-producing bacteria (right). (**C**) *Heritable exo-siRNAs silencing is reset by stress*. The graph displays the measured germline GFP fluorescence levels of individual worms (y-axis) across generations under the indicated condition (x-axis). Each dot represents the value of an individual worm. Shown are the median of each group, with box limits representing the 25th (Q1) and 75th (Q3) percentiles, notch representing a 95% confidence interval, and whiskers indicating Q1-1.5*IQR and Q3+1.5*IQR. FDR-corrected values were obtained using Dunn's test. (***) indicates q < 0.001 (see Materials and methods). The online version of this article includes the following figure supplement(s) for figure 1:

**Figure supplement 1.** Multiple durations and magnitudes of stress can induce resetting of small RNAs.

**Figure supplement 2.** Starvation and hyperosmotic stress do not affect the basal expression level of the GFP reporter.

**Figure supplement 3.** stress expedites the typical diminishment of heritable small RNA responses.

**Figure supplement 4.** Stress resets heritable small RNAs even when applied during adulthood.

**Figure supplement 5.** Resetting of heritable small RNA responses can only occur during the F1 generation.

suggesting a fundamental difference in the 'memory programs' of exogenous and endogenous transgenerational small RNA responses.

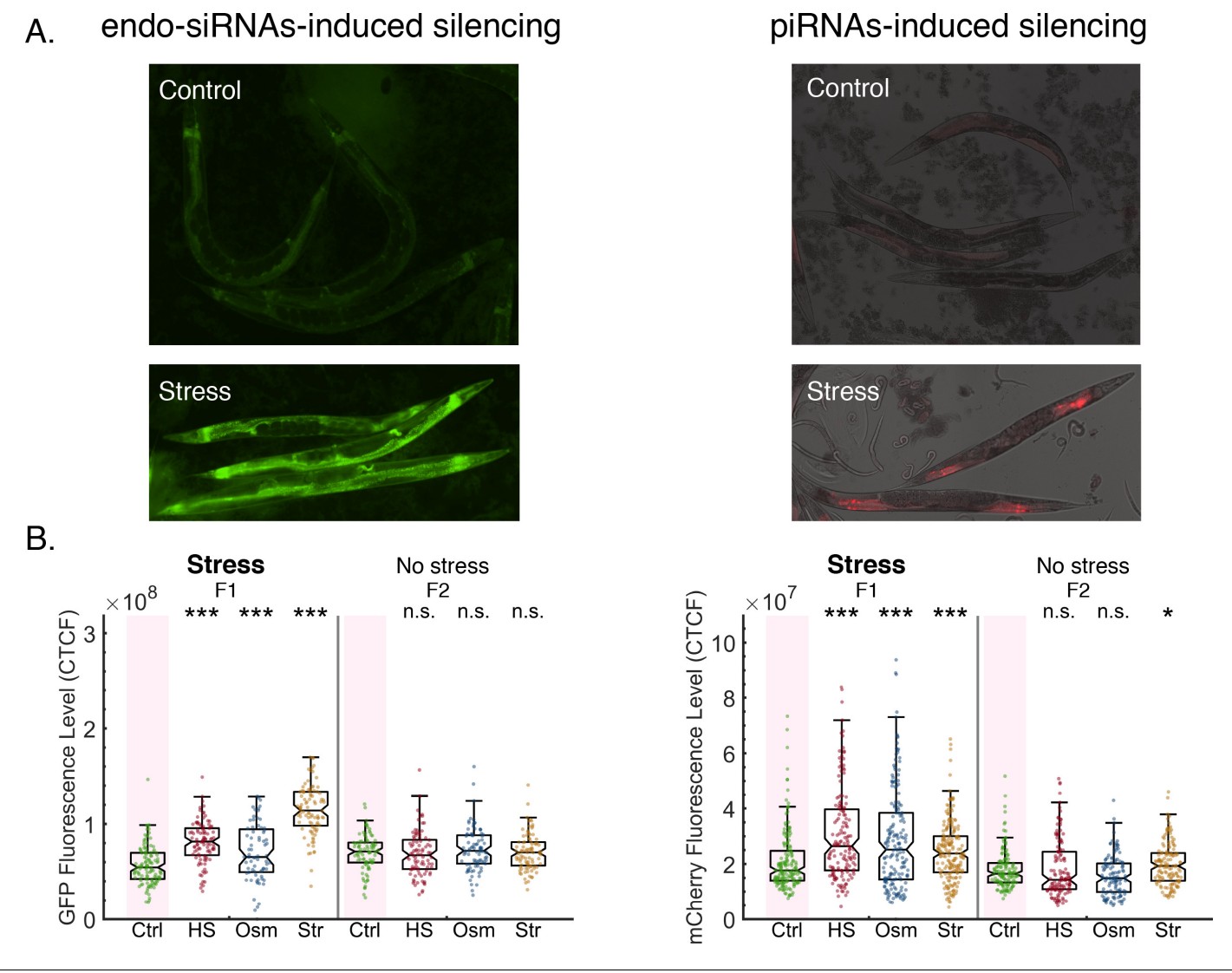

**Figure 2.** Stress resets endo-siRNAs and piRNAs-induced silencing. (**A**) Representative images of worms expressing endo-siRNAs sensor (left) or the piRNAs sensor (right) under control or stress (HS) conditions. (**B**) The graph displays the measured GFP (left, endo-siRNAs sensor) or mCherry (right, piRNAs sensor) fluorescence levels of individual worms (y-axis) across generations under the indicated condition (x-axis). Each dot represents the value of an individual worm. Shown are the median of each group, with box limits representing the 25th (Q1) and 75th (Q3) percentiles, notch representing a 95% confidence interval, and whiskers indicating Q1-1.5*IQR and Q3+1.5*IQR. FDR-corrected values were obtained using Dunn's test. Not significant (ns) indicates q ≥ 0.05, (*) indicates q < 0.05, and (***) indicates q < 0.001 (see Materials and methods).

## Stress, and not any change in growth conditions, leads to the resetting of ancestral small RNA silencing

Exposure to stress can be viewed simply as a change in regular growth conditions: from non-stressful to stressful settings. We next asked whether any form of change in growth conditions between the parental and the next generations could lead to the resetting of heritable small RNA responses. To this end, we examined how an 'improvement' in cultivation conditions across generations affects the dsRNA-derived heritable small RNA response. In this set of experiments, we induced a continuous stress at the parental generation by exposing the P0 worms, in which a heritable anti-*gfp* response is initiated, to a consistent but milder heat stress (25˚C, see Materials and methods). The next generation was then grown under either similar mild stress conditions or transferred to regular growth conditions (see scheme in *Figure 3A*). We found that resetting only occurs in response to stress (q-value <0.0001, *Figure 3A*); Progeny of stressed worms that were transferred to non-stressful

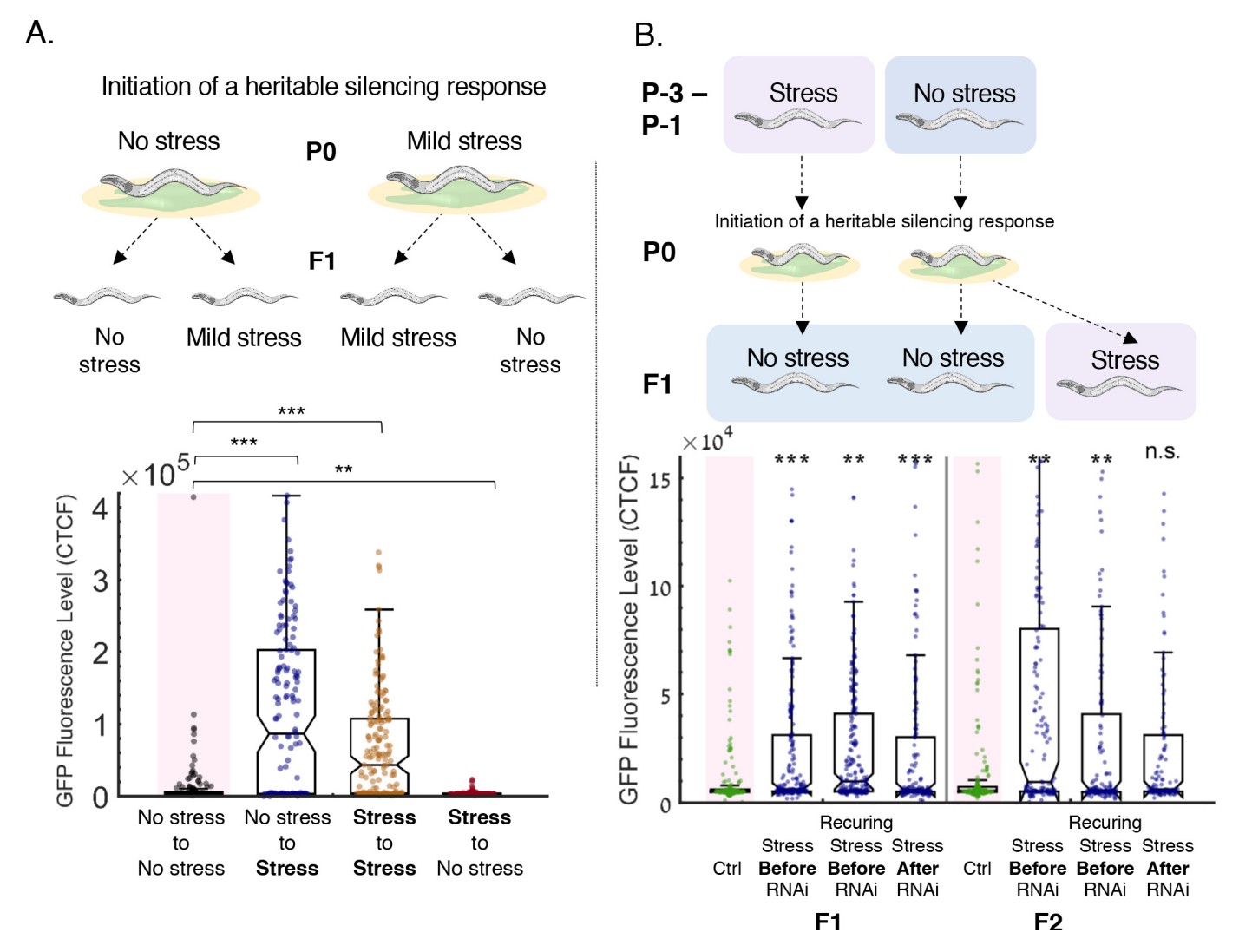

**Figure 3.** Resetting of heritable small RNA responses is induced specifically by stress, also when experienced prior to initiation of RNAi. (**a**) Shifting from stress to non-stress conditions fails to reset small RNA inheritance. Upper panel: *Experimental scheme.* Worms grown in regular growth conditions (20°C, control) or in high temperatures (25°C, stress) are exposed to an anti-*gfp* RNAi trigger. The next generation are then grown either in similar conditions (control to control, stress to stress) or transferred to the other growth condition (control to stress, stress to control). Lower panel: *Stress, and not any change in the environment, resets heritable small RNAs.* Worms which were exposed to high temperatures at the next generation reset the RNAi-induced *gfp* silencing regardless of the growth conditions at the previous generation. The graph displays the measured GFP fluorescence levels of individual worms (y-axis) under the indicated condition (x-axis). Each dot represents the value of an individual worm. Shown are the median of each group, with box limits representing the 25th (Q1) and 75th (Q3) percentiles, notch representing a 95% confidence interval, and whiskers indicating Q1-1.5*IQR and Q3+1.5*IQR. FDR-corrected values were obtained using Dunn's test. Not significant (ns) indicates q ≥ 0.05, (**) indicates q < 0.01 and (***) indicate q < 0.001 (see Materials and methods). (**b**) Worms exposed to stress prior to the initiation of RNAi also reset the heritable response. Upper panel: *Experimental scheme.* In the generations prior to the RNAi trigger, worms were either grown in regular growth conditions, starved for one generation, or starved for three consecutive generations (P-3 – P-1). The worms were then exposed to an RNAi trigger (P0). The F1 progeny of previously starved worms were grown in regular growth conditions ('*Stress Before RNAi*', '*Recurring Stress Before RNAi*'). The F1 progeny of previously unstressed worms were grown in either regular growth ('*Ctrl*') conditions or were starved ('*Stress After RNAi*'). All worms were then grown in regular growth conditions in the F2 generation. Lower panel: *Stress, when experienced prior to the initiation of RNAi, also resets the heritable response.* Worms which were exposed to starvation one or more generations before the initiation of RNAi reset the RNAi-induced *gfp* silencing in the next generations. The graph displays the measured GFP fluorescence levels of individual worms (y-axis) under the indicated condition (x-axis). Each dot represents the value of an individual worm. Shown are the median of each group, with box limits representing the 25th (Q1) and 75th (Q3) percentiles, notch representing a 95% confidence interval, and whiskers indicating Q1-1.5*IQR and Q3+1.5*IQR. FDR-corrected values were obtained using Dunn's test. Not significant (ns) indicates q ≥ 0.05, (**) indicates q < 0.01 and (***) indicate q < 0.001 (see Materials and methods).

conditions did not exhibit resetting of the heritable response. Moreover, two consecutive generations of mild stress led to a weaker inheritance compared to worms that were not exposed to stress at all (q-value <0.0001, *Figure 3A*), and RNAi responses initiated in stressed parents were strengthened when the progeny were transferred to non-stressful conditions (q-value <0.0053, *Figure 3A*). Overall, we conclude that stress, and not any change in growth conditions, leads to the resetting of heritable small RNA responses, even regardless of the ancestral growth conditions.

## Stress, when experienced prior to the initiation of RNAi, also reduces the heritable response

Distinct pools of endogenous and exogenous small RNA molecules compete over shared biosynthesis and amplification resources (*Gent et al., 2010*; *Lee et al., 2006*). Congruently, a reduction in function (e.g. through genetic manipulations) in one pathway can lead to an increase in function in the competing pathways (*Duchaine et al., 2006*; *Fischer et al., 2011*). Competition has also been indicated to regulate the transgenerational duration of heritable responses and underlie stable small RNA inheritance in multiple inheritance mutants (*Lev et al., 2019b*; *Lev et al., 2017*).

Previous works have shown that environmental perturbations, including the stress conditions we experiment with (starvation and heat stress), induce small RNA changes that persist across generations and create a 'transgenerational memory' of past experiences (*Ni et al., 2016*; *Rechavi et al., 2014*; *Schott et al., 2015*). Such transgenerational heritable responses following stress could in theory 'compete' over shared resources (*Duchaine et al., 2006*; *Fischer et al., 2011*; *Gent et al., 2010*; *Lee et al., 2006*) with other heritable responses and thus indirectly lead to a reduction – or resetting – of previously acquired inheritance programs. We therefore asked whether competition between different heritable small RNA programs could be involved in stress-induced resetting of parentally acquired responses. To test this possibility, we examined if stress that is applied *prior* to the initiation of the heritable dsRNA-induced response would affect the potency of inheritance in later generations. If stress only functions as a resetting signal within the same generation, we should not expect pre-exposure to stress to affect the heritable response that is initiated in later generations. Instead, we found that exposure to stress (starvation), even prior to the initiation of inheritance (for either one or three consecutive generations), leads to a similar resetting response as observed for stress that is applied in the next generation (*Figure 3B*). These results indicate that stress inheritance can in fact compete with other transgenerational responses, or, alternatively, affect the function of small RNA machinery in a transgenerational manner (see *Figure 3B* and Discussion).

## The effects of stress on the endogenous small RNA pools

Endogenous small RNAs align to large parts of the genome, and their inheritance is especially important for transgenerational regulation of germline expressed transcripts (*de Albuquerque et al., 2015*; *Gu et al., 2009*; *Phillips et al., 2015*). To better understand the global effects of stress-induced resetting on the worm's endogenous small RNA molecules, we sequenced small RNA from adult (day 1) worms which were exposed to stress at their first larval stage (heat, osmotic, and starvation stress) and from their progeny (see scheme in *Figure 4A* and Materials and Methods). Consistently with the observed phenotypic resetting of heritable responses following stress, in worms that were directly exposed to stress, all three types of stress conditions led to a reduction in the levels of heritable endogenous small RNA. However, across multiple repeats (see Materials and methods), different stressors shaped the small RNA pools of the next generation in different ways (*Figure 4B*): starvation stress led to a reduction in endogenous small RNA levels both in the worms that were directly exposed to stress and in their progeny. In contrast, heat stress caused a reduction in endogenous small RNA levels in the stressed generation, but most types of endogenous small RNA showed elevated levels in the next generation after heat stress (*Figure 4B*). We did not detect a consistent heritable change in the global levels of endogenous small RNAs following hyperosmotic stress (*Figure 4B*).

## Stress-induced resetting of target-specific small RNAs

The endogenous small RNA pathways in *C. elegans* vary greatly in the initial (or 'primary') source of small RNAs, the processing steps toward small RNA maturation, the co-factors that function in each pathway, their effects on gene expression, heritability potential, and the actual genes that they

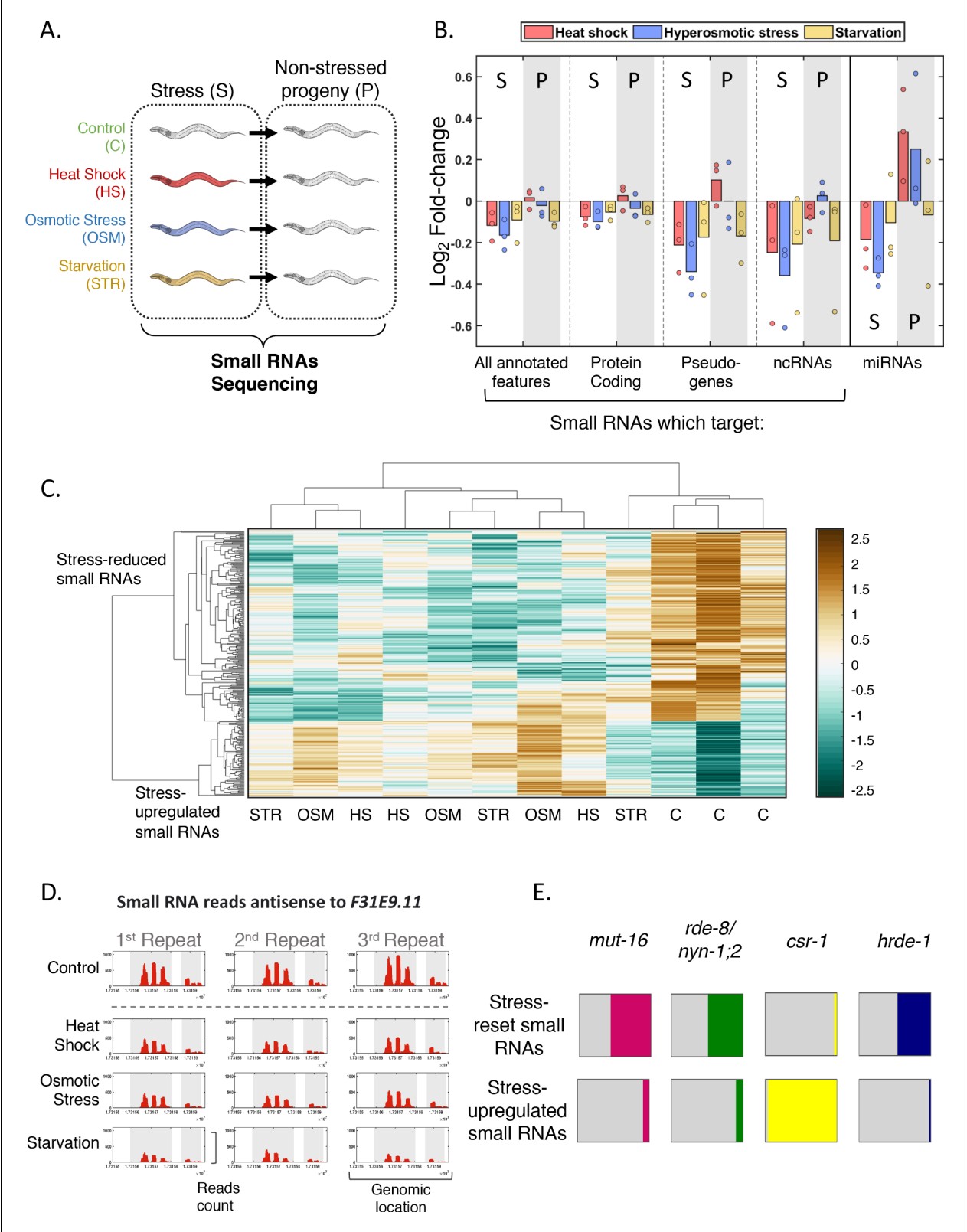

**Figure 4.** Genome-wide small RNA changes following stress. (**A**) Experimental scheme. Worms were exposed to stress during their first larval stage and were collected for RNA extraction and small RNA sequencing on the first day of adulthood. The next generation was grown under normal conditions (see Materials and methods). (**B**) Changes in total small RNA levels aligning to various genomic features. Presented are the Log₂ fold-change values (y-axis) for each condition (color coded) when compared to the control. Each dot represents one independent biological repeat.

*Figure 4 continued on next page*

*Figure 4 continued*

S = stressed generation. P = progeny (C) Clustering of the 281 stress-affected small RNA targets, based on their normalized total number of reads in each sample (four different conditions in three independent biological repeats). Hierarchical clustering was performed with Spearman's rank correlation as a distance metric and 'average' linkage. The data have been standardized across all columns for each gene, so that the mean is 0 and the standard deviation is 1. (D) *An example of a stress-affected small RNAs target.* The *F31E9.11* gene is covered by small RNAs which are reset across all stress conditions. Shown are the normalized read counts (y-axis) as function of genomic location (x-axis) of small RNAs targeting the *F31E9.11* gene. Exons appear on a gray background. (E) *Overlap of targets of stress-affected small RNAs with known targets of different small RNA pathways.* Each square represents the proportional overlap of reset (upper row) or upregulated (bottom row) small RNA targets with known targets of the indicated small RNA pathways. Shown are results for MUT-16-dependent small RNA targets (*Zhang et al., 2011*), NYN-1;2/RDE-8-dependent small RNA targets (*Tsai et al., 2015*), CSR-1-bound small RNA targets (*Claycomb et al., 2009*) and HRDE-1-bound small RNA targets (*Buckley et al., 2012*).

target (*McMurchy et al., 2017*). To better explore how stress-induced resetting affects each pathway, we examined stress-induced changes in small RNA regulation over specific genes and identified a list of 281 genes that were targeted by stress-affected endogenous small RNAs (regardless of the stress type). FDR < 0.1, *Figure 4C and D*, *Supplementary file 1* and Materials and methods). Interestingly, the targets of these stress-affected small RNAs show enrichment for stress-related functions. As a group, these genes were shown to be regulated by the TGF-beta pathway and the insulin pathway, and to be affected in response to rapamycin exposure and changes in temperature (*Calvert et al., 2016*; *Chen et al., 2015*; *Shaw et al., 2007*; *Viñuela et al., 2011*).

In agreement with the observed overall reduction in endogenous small RNA levels and the phenotypic resetting of the silencing responses, in most of the gene targets of stress-affected small RNAs (202/281, 72%) the targeting small RNAs were down-regulated, or 'reset', following stress. These endogenous small RNAs exhibited a strong enrichment for *hrde-1*-dependent small RNAs and *Mutator*-dependent small RNAs (*Phillips et al., 2014*; *Yang et al., 2016*; *Zhang et al., 2011*) (see *Figure 4C and E*). The HRDE-1 (**H**eritable **R**NAi **De**ficient-1) Argonaute is an important regulator of transgenerational gene silencing by small RNAs and, while largely dispensable for silencing within the same generation, is explicitly required for transmission of silencing responses across generations (*Ashe et al., 2012*; *Buckley et al., 2012*; *Shirayama et al., 2012*). The *Mutator* proteins (*mut-16*, *mut-14;smut-1 Phillips et al., 2014*; *Zhang et al., 2011*) were shown to be involved in multiple endogenous small RNA biogenesis pathways, affecting both somatic and germline small RNAs, and are required for efficient small RNA amplification and cleavage of target RNAs (*rde-8* and *nyn-1;nyn-2 Tsai et al., 2015*). Generating (via crossing) *mut-14;smut-1;mut-16* triple mutants was previously shown to enable erasure of heritable small RNA-based memory of self and non-self-genes (*de Albuquerque et al., 2015*; *Phillips et al., 2015*). Additionally, we found here that targets of stress-reduced small RNAs are enriched for dsRNA-producing loci (*Saldi et al., 2014*) and piRNAs gene targets (*Bagijn et al., 2012*) (see the full list of enrichments in *Supplementary file 3*).

In contrast to the enrichment of HRDE-1 and *Mutator*-dependent small RNAs among small RNAs which are reset following stress, the targets of small RNAs that showed elevated levels of small RNAs following stress (79/281, 28%) were almost exclusively CSR-1-dependent endo-siRNAs (hypergeometric test, p-value=2.9 e−48, 96% were found to be physically bound to CSR-1, *Claycomb et al., 2009*; *Figure 4E*). Unlike other endo-siRNAs pathways, endo-siRNAs which are bound by the CSR-1 Argonaute were demonstrated to promote gene expression rather than gene silencing in the worms (*Wedeles et al., 2013*).

Finally, we did not detect widespread changes in stress-affected small RNAs in the progeny of stressed worms, when accounting for all stress types together (only 10 genes were targeted by such small RNAs, *Supplementary file 2*), in accordance with the varying and stress-dependent global changes in small RNA levels in the next generation after stress.

## Stress represses the expression of multiple small RNA factors

To look for potential effectors of stress-induced resetting of small RNAs, we examined mRNA-sequencing data from worms under heat shock, hyperosmotic stress, or starvation (*Dodd et al., 2018*; *Finger et al., 2019*; *Schreiner et al., 2019*). We found that multiple epigenetic, small RNA, and p-granules factors consistently show significantly reduced gene expression levels following stress, regardless of the stress types (73 epigenetic-related genes significantly downregulated in all stress conditions with FDR < 0.1, *Supplementary file 4*, hypergeometric test; fold-enrichment = 2.4,

adjusted p-value<0.00001). Specifically, we found that the Argonaute genes *hrde-1*, *rde-1*, *ergo-1*, *nrde-3*, *wago-1,* and *alg-2* are all significantly downregulated across all stress types. These Argonautes were previously shown to regulate and be necessary for RNAi responses both within and across generations (*Buckley et al., 2012*; *Billi et al., 2014*; *Grishok et al., 2001*). Additionally, we found that the *Mutator* genes *mut-2* and *mut-16*, whose small RNAs were depleted following stress, and the factors *rde-4*, *rrf-3*, *hrde-4,* and *rde-8* all show significantly reduced gene expression levels in response to stress (*Supplementary file 4*). These epigenetic factors were previously found to be required for the synthesis, amplification, and inheritance of small RNAs across generations (*Duchaine et al., 2006*; *Lee et al., 2006*; *Parker, 2006*; *Spracklin et al., 2017*; *Tabara et al., 2002*; *Tsai et al., 2015*). Overall, we find that stress induces a widespread suppression of various components of the small RNA machinery, in accordance with the observed global resetting of small RNA responses and multiple types of endogenous small RNAs (*Figure 4*).

## The p38 MAPK pathway regulates small RNA resetting in response to stress

As resetting of small RNAs seems to be induced specifically by stress, we next examined whether the different and well-characterized stress pathways in the worm could also regulate small RNA resetting. To this end, we tested the ability of multiple mutants that are defective in various stress pathways to reset small RNA inheritance in response to stress. Since we observed similar stress-induced resetting for multiple types of stress conditions, we chose to examine mutations in genes that function as 'hubs' of multiple stress signaling pathways. Namely, genes that regulate the worm's stress response in a stress-type independent manner.

Interestingly, out of the 11 mutants defective in stress responses that we examined, eight mutants showed a general enhanced (*pmk-1*/p38 MAPK, *sek-1*/p38 MAP2K, *skn-1*/Nrf2, *daf-2*/InsR) or defective (*mek-1;sek-1*, *kgb-1*, *daf-2*/InsR;*daf-16*/FOXO, *hsf-1*/HSF1) RNAi inheritance (*Figure 5A* and *Figure 5—figure supplements 1* and *2*). This was true irrespectively of whether the mutant worms experienced stress or not and suggests that stress regulation and small RNA inheritance are interconnected even in the absence of stress.

We then examined the direct involvement of these genes in resetting heritable RNAi in response to stress by initiating a heritable anti-*gfp* dsRNA-derived silencing response in these mutants and exposing the next generation to three types of stress. We found that MAP Kinase (MAPK) genes (*sek-1*/*mek-1*, *sek-1*, *pmk-1*, *kgb-1*), and the *skn-1* gene are required for resetting of RNAi inheritance in response to stress (*Figure 5B and C*, and *Figure 5—figure supplements 1* and *2*), in addition to their effects on RNAi inheritance even in the absence of stress, as described above. *skn-1*/Nrf2 encodes a transcription factor which is regulated by p38 MAPK-dependent phosphorylation (*Inoue et al., 2005*). Importantly, all the examined mutant worms expressed the GFP transgene in similar levels to wild-type worms in the absence of anti-*gfp* RNAi response (*Figure 5—figure supplement 3*).

We recently showed that HSF-1 activity corresponds to the 'inheritance state' of worms when initiating a heritable RNAi response (*Houri-Zeevi et al., 2020*). This 'inheritance state' determines the fate of the heritable response (its persistence across generations). Our finding that *hsf-1* mutants are capable of resetting RNAi inheritance in response to stress suggests that HSF-1's role is likely in the initiation, and not the maintenance, of heritable RNAi responses.

The MAPK pathway is necessary for the integration of responses to multiple types of stress, such as DNA damage (*Bianco and Schumacher, 2018*; *Ermolaeva et al., 2013*), osmotic stress (*Gerke et al., 2014*), heat shock (*Mertenskötter et al., 2013*), and pathogen infection (*Troemel et al., 2006*). Interestingly, we found that the ability to reset heritable responses following stress, and the ability to transmit RNAi, appear to be two distinct functions: some of the mutants that did not reset heritable silencing following stress showed enhanced RNAi inheritance (*sek-1*, *pmk-1*, *skn-1*, acting in the same signaling cascade), while others were defective in RNAi inheritance (*mek-1*/*sek-1* and *kgb-1*) (*Figure 5A*). Overall, we conclude that resetting of heritable silencing following stress is a regulated process that is mediated by the MAPK pathway and by the *skn-1* transcription factor.

To further explore the regulatory role of the identified stress-resetting factors in small RNA inheritance, we examined the tissue-specific requirements of the MAPK pathway in stress-induced resetting. *sek-1*, which is required for stress-induced resetting of small RNAs, was shown to be expressed

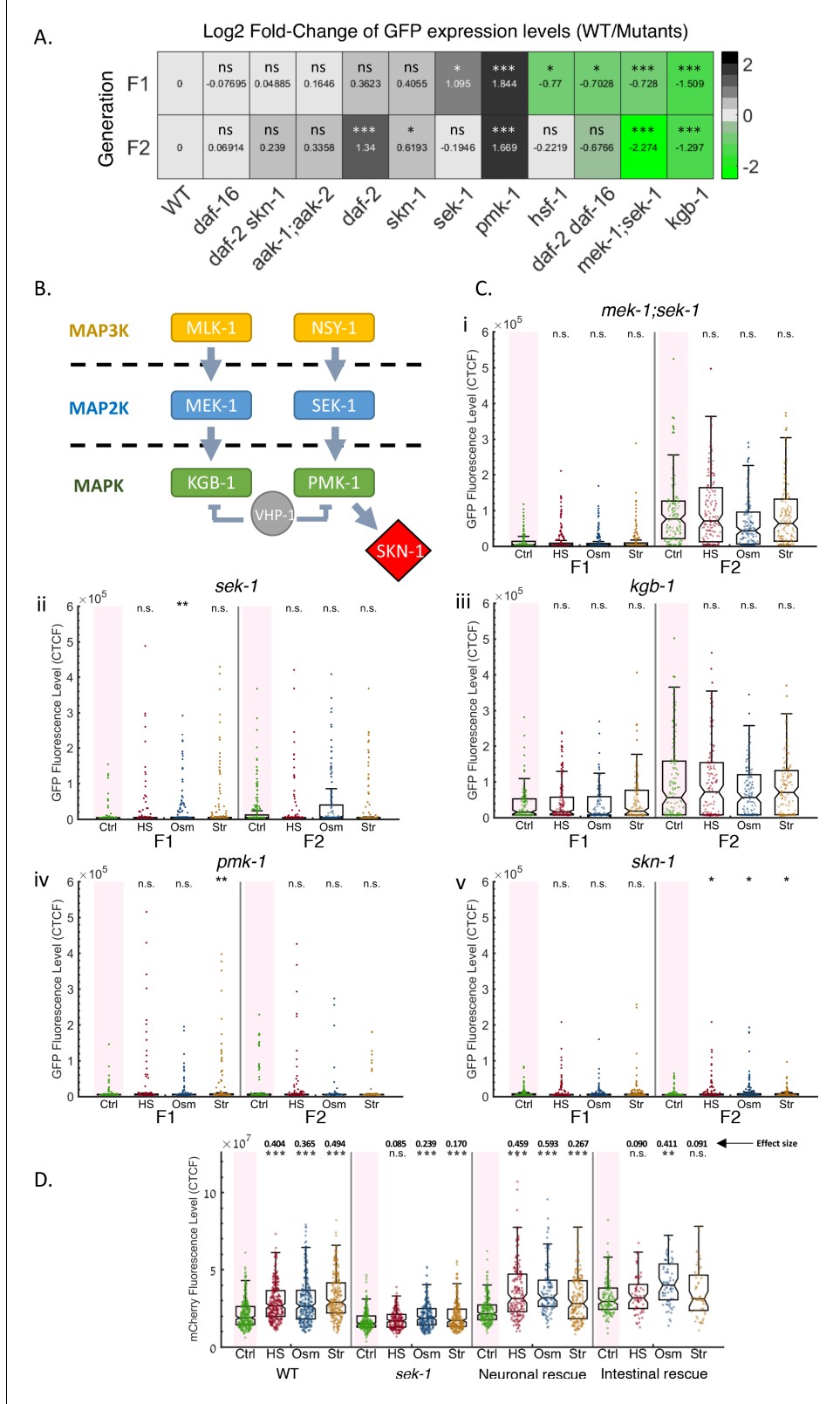

**Figure 5.** The MAPK pathway and the SKN-1 transcription factor regulate small RNAs resetting in response to stress. (**A**) Mutants defective in stress-responsive genes display altered heritable *RNAi dynamics*. Heatmap representing the $\log_2$-fold change of GFP fluorescence levels (color coded and indicated values) in mutants worms compared to WT at the F1 (upper panel) and F2 (lower panel) generations after RNAi. Shown are results under normal conditions (no stress). FDR-corrected values were obtained using Dunn's test. The comparison values of each mutant were calculated based on

*Figure 5 continued*

its independent experiments (see full results in *Figure 5—figure supplements 1* and *2*). (B) Members of *C. elegans'* MAPK pathway, found to influence resetting of small RNAs. Based on *Ewbank, 2006*; *Inoue et al., 2005*; *Kim et al., 2004*; *Mizuno et al., 2008* (C) *Mutants defective in the MAPK signaling pathway do not reset heritable RNAi responses following stress.* The graphs display the measured germline GFP fluorescence levels of mutant worms (y-axis) across generations under the indicated condition (x-axis). Each dot represents the value of an individual worm. Each mutant was examined in an independent experiment. Shown are the median of each group, with box limits representing the 25th (Q1) and 75th (Q3) percentiles, notch representing a 95% confidence interval, and whiskers indicating Q1-1.5*IQR and Q3+1.5*IQR. FDR-corrected values were obtained using Dunn's test. Not significant (ns) indicates $q \geq 0.05$, (*) indicates $q < 0.05$, (**) indicates $q < 0.01$, and (***) indicates $q < 0.001$, (see Materials and Methods). Full results (including the side-by-side wild-type results of each experiment) can be found in *Figure 5—figure supplements 1* and *2*. (D) *Neuronal rescue of sek-1 is sufficient to restore stress-induced resetting of piRNAs-induced silencing.* The graph displays the measured mCherry fluorescence levels of individual worms (y-axis) across generations under the indicated condition (x-axis). Each dot represents the value of an individual worm. Shown are the median of each group, with box limits representing the 25th (Q1) and 75th (Q3) percentiles, notch representing a 95% confidence interval, and whiskers indicating Q1-1.5*IQR and Q3+1.5*IQR. FDR-corrected values were obtained using Dunn's test. Not significant (ns) indicates $q \geq 0.05$, (**) indicates $q < 0.01$, and (***) indicates $q < 0.001$ (see Materials and methods). Effect size estimates (shown above the asterisks) were obtained using the Cliff's Delta estimate (see Materials and methods).

The online version of this article includes the following figure supplement(s) for figure 5:

**Figure supplement 1.** Multiple stress signaling and processing pathways affect heritable RNAi dynamics and stress-induced resetting of heritable silencing.

**Figure supplement 2.** Mutants of the *daf-16, aak-1/2* and *daf-2* genes are capable of stress-induced resetting of heritable responses.

**Figure supplement 3.** Mutants in the MAPK pathway, the transcription factor SKN-1, and the putative H3K9 methyltransferase MET-2 do not affect the basal expression of the GFP reporter.

in neurons and in the intestine (REFs), two tissues which have a role in sensing and communicating environmental perturbations. We therefore used strains carrying a neuronal rescue or an intestinal rescue of the *sek-1* gene (*Shivers et al., 2009*) (see Materials and methods). Due to GFP expression which is inherent to these tissue-specific rescue strains (see Materials and methods and Key Resources Table), we chose to study the effects of neuronal and intestinal rescues of *sek-1* on the silencing of the piRNA sensor (mCherry fluorescence) following stress. While this sensor provides indication of endogenous small RNAs silencing, we still observed that both exogenous and endogenous silencing responses are subjected to resetting following stress within the same generation (as described in the previous sections). Moreover, the rescue experiments were performed side by side with mutant lines that enabled a direct comparison of the general effects of the examined mutations on resetting of endogenous silencing following stress. These experiments included a particularly large quantity of worms tested over multiple biological and technical replicates over many conditions (N = 3193). This led to increased sensitivity during hypothesis testing and the detection of statistically significant effects that may not be biologically significant. We therefore proceed to report an effect size for each group in addition to corrected p-values (Cliff's Delta effect size measure; higher absolute value indicates a larger effect size. See Materials and methods) (*Cliff, 1993*).

We found that neuronal expression of *sek-1* is sufficient to re-establish a strong resetting of silencing in response to stress (*Figure 5D*, Cliff's Delta = 0.4597, 0.593, 0.2672 for heat stress, hyperosmotic stress, and starvation stress, respectively). Intestinal rescue of *sek-1*, on the other hand, did not regenerate a strong resetting response following heat stress and starvation stress (Cliff's Delta = 0.0905, 0.0909), but did regenerate a strong resetting response following osmotic stress (Cliff's Delta = 0.4108). However, we note that the general levels of silencing in worms that carry an intestinal rescue of *sek-1* were reduced even in the absence of stress and thus we cannot exclude that stress-induced resetting in these worms is masked by generally low levels of silencing (*Figure 5D*). Notably, osmotic stress is the only stress condition that did not show a consistent effect on the total small RNA pools in unstressed progeny (*Figure 4B*). Moreover, unlike starvation and heat stress, osmotic stress was shown to elicit an intergenerational, and not a transgenerational, epigenetic response (*Burton et al., 2017*; *Rechavi et al., 2014*; *Schott et al., 2015*), that is mediated through the intestine. It is thus plausible that osmotic stress affects resetting via the intestine as well, unlike the other stress conditions.

Overall, we conclude that while many stress-related factors in the worm affect the basic function of the RNAi inheritance machinery (*Figure 5A*), the p38 MAPK pathway – and the SKN-1/Nrf2 transcription factor which is regulated by it – are specifically required for resetting of small RNAs in response to stress.

## SKN-1/Nrf2 regulation over small RNA factors

Among the stress-related factors that we found to be required for stress-induced resetting of small RNAs, SKN-1 is the only transcription factor. It is therefore plausible that SKN-1 could function in stress-induced resetting of small RNAs by directly regulating the expression of various epigenetic and small RNA factors. Moreover, recent findings have implicated SKN-1 in transgenerational inheritance in *C. elegans* (*Burton et al., 2020*; *Ewe et al., 2020*). We therefore sought to elucidate the regulatory relationship between SKN-1 and small RNA factors.

To that end, we inspected mRNA-sequencing data from worms that underwent anti-*skn-1* RNAi (*Dodd et al., 2018*; *Michael et al., 2015*). In particular, we examined genes that are normally repressed in response to stress but show significantly weaker repression in anti-*skn-1* RNAi background (see Materials and methods). We found that this group of genes is significantly enriched for epigenetic genes and p-granule factors (599 genes, 35 of which are epigenetic-related genes; q-values = 0.000134, 0.00680, *Supplementary files 5* and *6*). In particular, we found that the Argonaute gene *nrde-3* and the epigenetic factors *rrf-3*, *rde-8*, and *nyn-2* are downregulated in response to stress in a *skn-1*-dependent manner, in accordance with the epigenetic factors we found to be downregulated in response to all stress conditions (*Supplementary file 4*).

To identify other potential downstream effectors of *skn-1* in stress-induced resetting of small RNAs, we examined the putative promoter regions of known epigenetic genes for the binding motif sequence of SKN-1. Among the epigenetic genes whose promoter regions contained SKN-1 binding sites (63 genes) were the putative RNA-dependent RNA polymerase *rrf-3* and the putative H3K9 histone methyltransferase *met-2* (p-value=0.0001, 0.0001, *Supplementary file 7*; *Pujato et al., 2014*). Overall, we conclude that SKN-1 could affect the expression of multiple epigenetic and small RNA factors, including factors that were previously shown to regulate the duration of heritable RNAi responses.

## Discussion

In this study, we found that stress leads to the resetting of transgenerationally transmitted small RNAs, erasing heritable gene regulatory responses. Stress-induced resetting depends on the p38 MAPK pathway and the SKN-1 transcription factor, and is regulated by MET-2, a putative H3K9 methyltransferase that is required for germline reprogramming (*Kerr et al., 2014*). We found that stress induces a reduction in small RNA levels across the genome. This reduction stems from changes in multiple distinct small RNA species, including small RNAs that regulate stress-related genes. We speculate that a mechanism for resetting of heritable small RNAs could be adaptive in rapidly changing environments (see model in *Figure 6C*).

We previously described a tunable mechanism that controls the duration of heritable RNAi responses and found that ancestral RNAi responses can be enhanced by non-target-specific reactivation of the RNAi system in the progeny (*Houri-Ze'evi et al., 2016*). We hypothesized that when both parents and progeny are challenged by dsRNA-induced RNAi it could be beneficial for the worms to extend the duration of ancestral RNAi responses. We reasoned this could be the case since under these circumstances the progeny's environment resembles the parent's, and therefore the heritable response could still be relevant for the progeny. In contrast, stress-induced resetting of heritable small RNAs, the phenomenon described in this manuscript, could unburden descendants from heritable information that is no longer relevant.

Accumulating evidence from the past two decades, collected in different organisms, suggests that stress can lead to intergenerational and transgenerational epigenetic changes (*Bohacek and Mansuy, 2015*). In certain instances, stress was found to lead to a transient heritable reduction in small RNAs and chromatin marks (*Belicard et al., 2018*; *Klosin and Lehner, 2016*). Such heritable effects on chromatin were referred to as 'epigenetic wounds' which 'heal' following gradual re-accumulation of the marks over generations (*Klosin and Lehner, 2016*). Our results suggest that reduction in small RNAs following stress is an active and regulated process that is initiated by stress-signaling pathways (and specifically, the MAPK pathway). Additionally, it was previously shown that parental stress can provide the progeny with survival advantage in the face of additional stress (*Kishimoto et al., 2017*). It would be interesting to explore whether resetting of small RNAs in response to stress can serve as a transgenerational signal that 'primes' the progeny for additional stress periods.

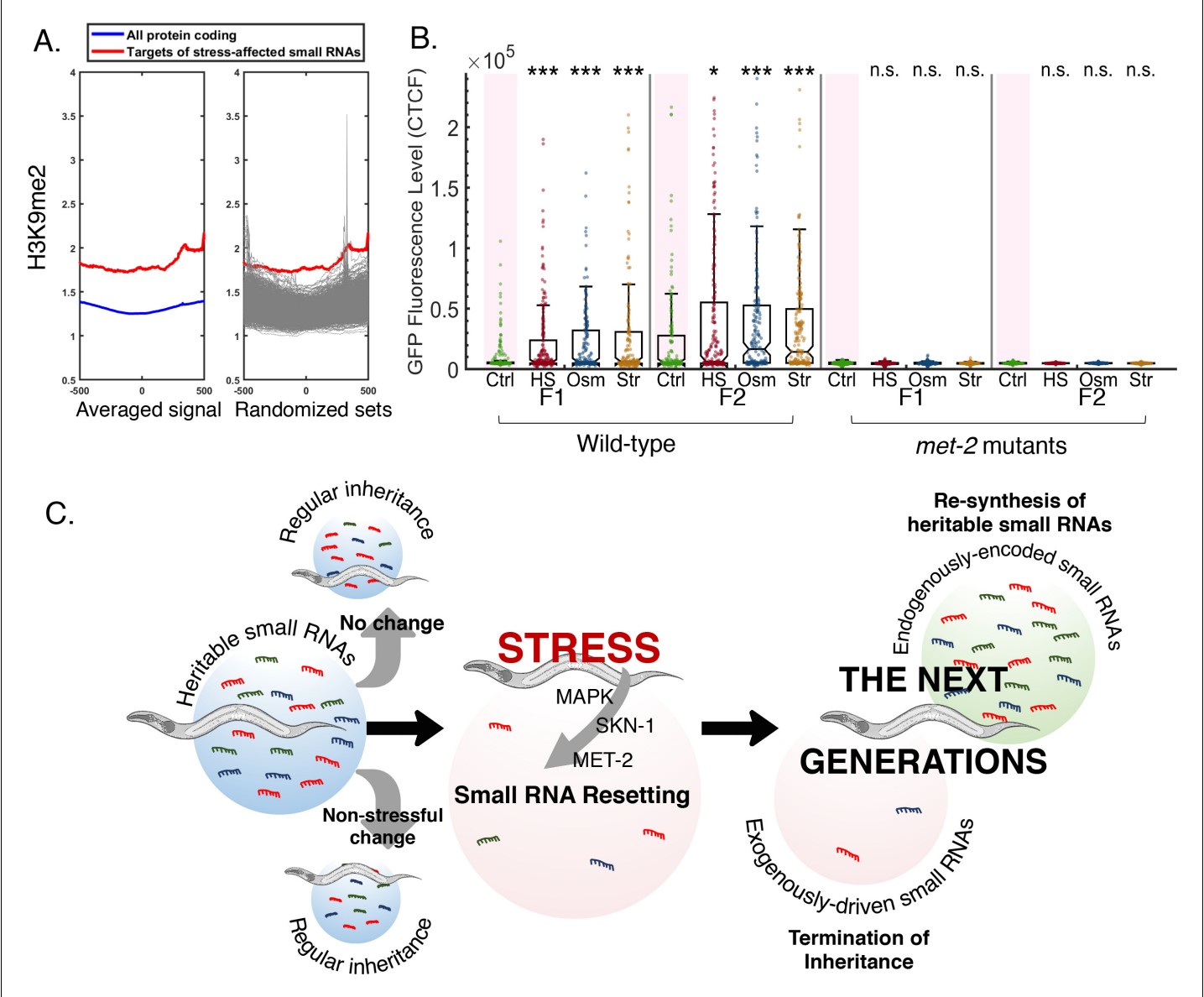

**Figure 6.** Stress-induced small RNA resetting depends on the H3K9 methyltransferase MET-2. (**A**) *Targets of stress-affected small RNAs show significantly increased H3K9me2 marks*. An analysis of *H3K9me2* signal (based on published data from ***McMurchy et al., 2017***). Presented is the averaged H3K9me2 signal (y-axis) of all protein coding genes (blue) and target genes of stress-affected small RNAs (red). All genes are aligned according to their Transcription Start Sites (TSS), and the regions of 500 base pairs upstream and downstream of the TSS are shown on the x-axis. Each gray line (right panel) represents the average of a random set of genes (500 iterations) equal in size to the set of target genes of stress-affected small RNAs. (**B**) *met-2 mutant worms do not reset heritable small RNAs in response to stress*. The graph displays the measured germline GFP fluorescence levels of wild-type and *met-2* mutant worms (y-axis) across generations under the indicated condition (x-axis). Each dot represents the value of an individual worm. Shown are the median of each group, with box limits representing the 25th (Q1) and 75th (Q3) percentiles, notch representing a 95% confidence interval, and whiskers indicating Q1-1.5*IQR and Q3+1.5*IQR. FDR-corrected values were obtained using Dunn's test. Not significant (ns) indicates q ≥ 0.05, (*) indicates q < 0.05, and (***) indicates q < 0.001 (see Materials and methods). (**C**) *A model summarizing stress-induced resetting of heritable small RNAs*. Small RNAs from both endogenous and exogenous sources are reset in response to stress. Resetting is mediated by the MAPK pathway, the SKN-1 transcription factor and the H3K9 methyltransferase MET-2. Endogenous small RNAs which are encoded in the genome are re-synthesized in the next generations, while small RNAs from exogenous sources and transient responses are eliminated.

The online version of this article includes the following figure supplement(s) for figure 6:

**Figure supplement 1.** Targets of stress-affected small RNAs show unique H3K9 marks and the H3K9 methyltransferase *met-2* is required for the execution of small RNA resetting.

Interestingly, we find here that stress that is applied prior to the initiation of RNAi inheritance leads to reduced heritable response (*Figure 3B*). This suggests that direct 'erasure' of the current pools of small RNAs also shapes small RNA inheritance in later generations, perhaps through competition between exogenous and endogenous small RNA pathways and its role in tuning the duration of small RNA inheritance (*Houri-Ze'evi et al., 2016*).

Active removal of epigenetic regulation in response to stress could serve as a mechanism for increasing genetic and phenotypic variability. Small RNAs in *C. elegans* distinguish between self and non-self-genes (*Shirayama et al., 2012*), and orchestrate gene expression in the germline and during development (*Feng and Guang, 2013*; *Gu et al., 2009*; *Han et al., 2009*). Relieving the regulation of heritable small RNAs could be a way to increase phenotypic plasticity or to reveal hidden genetic variability. For example, a conserved microRNA in flies was shown to buffer developmental programs against variation (*Li et al., 2009*).

We have recently found that changes in neuronal small RNA levels generate transgenerational effects (*Posner et al., 2019*). Similarly, it has been shown that olfactory memory can become heritable (*Moore et al., 2019*; *Pereira et al., 2019*; *Remy, 2010*). The systemic response to stress that KGB-1, SKN-1, and SEK-1 mediate (factors which are shown here to be required for resetting) was shown in the past to be coordinated by the nervous system (*Bishop and Guarente, 2007*; *Liu et al., 2018*; *Shivers et al., 2009*). Moreover, our results suggest that *sek-1* regulates stress-induced resetting of small RNAs through the worms' nervous system. Therefore, it could be interesting to understand if and how resetting of heritable small RNAs is controlled by neuronal activity.

In previous studies, it was shown that MET-2, a putative histone methyltransferase required for mono- and di-methylation of H3K9 and the homologue of mammalian SETDB1, is essential for termination of RNAi inheritance and establishment of an epigenetic 'ground state' (*Garrigues et al., 2015*; *Greer et al., 2014*; *Kerr et al., 2014*; *Lev et al., 2017*; *Mutlu et al., 2018*; *Towbin et al., 2012*), and that in *met-2* mutants aberrant endo-siRNAs accumulate over generations, eventually leading to sterility (Mortal Germline, Mrt phenotype) (*Andersen and Horvitz, 2007*; *Lev et al., 2017*; *Yang et al., 2019*).

Examination of published datasets revealed that targets of stress-affected small RNAs were enriched for genes which were previously found to be misregulated in *met-2* mutants (FDR = 2.1e-05) (*Zeller et al., 2016*) and that these genes have significantly higher than expected levels of H3K9me2 (*Figure 6A*, *Figure 6—figure supplement 1* and Materials and methods). Furthermore, we observe that *met-2*'s expression levels are significantly downregulated in worms under heat stress, osmotic stress, and starvation stress (q-values = 0.002211, 0.048440, 0.048440). Interestingly, we also observed that MET-2 is essential for stress-induced resetting of heritable RNAi (*Figure 6B*), even when accounting for the exceptionally strong RNAi inheritance in *met-2* mutants. *met-2* mutants were resistant to small RNAs resetting even when weaker RNAi responses were induced and did not show any alteration of the heritable RNAi responses following stress (*Figure 6—figure supplement 1B*). Overall, MET-2 seems to be required for the execution of resetting of heritable small RNA responses following stress. However, future work will determine the potential connection between MET-2 and stress regulation in the worms, and how these might synergize to generate stress induced small RNA resetting.

Non-DNA-based inheritance could be involved in multiple human disorders (*Crews et al., 2014*; *Nilsson et al., 2018*; *Tucci et al., 2019*). It is still unclear whether the same mechanisms that allow transgenerational inheritance in worms exist in mammals (*Horsthemke, 2018*). If analogous mechanisms are indeed conserved, understanding the pathways that counteract transmission or maintenance of heritable small RNAs could aid in prevention or treatment of various diseases.

## Materials and methods

### Experimental procedures

#### Worms' maintenance

Standard culture techniques were used to maintain the nematodes. The worms were grown on Nematode Growth Medium (NGM) plates and fed with OP50 bacteria. Except when otherwise noted, all experiments were performed at 20℃. The worms were kept fed for at least five generations before the beginning of each experiment. Extreme care was taken to avoid contamination or starvation.

Contaminated plates were discarded from the analysis. All experiments were performed with at least three biological replicates. The nematode strains used in this study are indicated in the Key Resources Table.

## RNAi treatment

The standard assay for RNAi by feeding was carried out as previously described (*Kamath et al., 2000*): HT115 bacteria that transcribe dsRNA targeting *gfp*, or control HT115 bacteria that only contain an empty vector plasmid, were grown in LB containing 100 ng/ml Carbenicillin, and were then seeded on NGM plates that contain 100 ng/ml Carbenicillin and 1mM IPTG . In the generation of worms that were exposed to RNAi (P0), worms were cultivated on those plates.

In experiments where the concentration of RNAi bacteria was controlled, a spectrophotometer was used to measure optical density at OD600. HT115 bacteria were diluted with empty vector bacteria to 2 OD and 0.5 OD.

## Calibration of stress conditions

For each stress condition, multiple durations and/or intensities of stress were tested (see *Figure 1—figure supplement 1*): for heat shock, worms; for osmotic stress, worms; and for starvation, worms were starved for either 3 days or 6 days.

The chosen conditions for each stress, which are described below, were chosen to be the longest/most intense conditions that most of the worm population would survive. This was done in order to avoid potential epigenetic selection that would bias the mean small RNA inheritance of the stressed population.

## L1 stress

Synchronized eggs were obtained by bleaching gravid adult worms. The progeny was then subjected to the various stress conditions.

## Heat shock

Heat shock treatment was performed at 37°C. The nematodes were heat-shocked 24 hr after the bleach treatment, for 120 min. After heat shock, the treated worms were placed back in an incubator set to 20°C, with each plate placed in a separate location (not in a pile) to avoid temperature differences between the top/bottom plates and the middle plates.

## Hyperosmotic stress

After bleach, the obtained eggs were seeded on high-salt growth media plates (350 mM NaCl) and were grown on these plates for 48 hr. After 48 hr, the worms were washed off with M9 buffer onto regular NGM plates.

## Starvation

For starvation conditions, the nematodes were grown on empty NGM plates and were maintained this way for a total of 6 days. After 6 days, the worms were washed off with M9 onto regular NGM plates seeded with OP50 bacteria. In the experiment with *aak-1/2* double mutants, both the wild-type and mutant worms were starved for 3 days instead of 6 days, due to starvation-sensitivity of the mutants.

In the experiments described in *Figure 3A* (a reverse change in settings; from stressed conditions to regular growth conditions), P0 nematodes were grown to adulthood on plates with RNAi bacteria in either a 20°C or a 25°C. The nematodes were bleached in adulthood, and their progeny were grown to adulthood on NGM plates either in a 20°C or a 25°C.

In the experiments described in *Figure 3B* (stress applied before exposure to RNAi), worms were exposed to either three consecutive generations of starvation (P-3 – P-1), one generation of starvation (P-1), or no starvation. P-1 mothers from the three conditions were allowed to lay eggs on plates with RNAi bacteria, and their progeny (P0) were grown to adulthood on said plates. The P0 mothers were bleached in adulthood, and their progeny (F1) were grown on NGM plates. Half of the non-starved F1 worms were grown on plates with food, and the other half were starved for 6 days.

## Adults stress

In the adult stress assays (*Figure 1—figure supplement 4*), stress conditions were applied when the worms reached young adulthood, with heat shock lasting 1 hr, hyperosmotic stress lasting 24 hr, and starvation lasting 48 hr.

## The next generations

In both assays (L1 and adults stress), the next generation of worms was cultured by randomly picking several mothers from each plate to a fresh plate and allowing them to lay eggs for few hours (8–24 hr). The number of picked worms and the length of the egg-laying period were constant within each experiment across all plates and conditions.

## piRNAs- and endo-siRNAs-derived silencing assays

To obtain expression of the piRNAs-silenced mCherry and the endo-siRNAs-silenced GFP, worms were grown in 25°C for three generations and then transferred back to recovery at 20°C. After two generations, worms were bleached and assayed for the different stress conditions, as described above.

## Tissue-specific rescue assays

We used strains ZD202 and ZD193, which both carry the *sek-1(km4)* mutation. Additionally, ZD202 carries a neuron-specific rescue of *sek-1* tagged with GFP (*unc-119p::sek-1*(cDNA)::GFP::*unc-54*–3' UTR), and ZD193 carries an intestine-specific rescue of *sek-1* tagged with GFP (*ges-1p::sek-1* (cDNA)::GFP::*unc-54*–3' UTR) (see Key Resources Table). Due to the inherent GFP expression of these tagged rescue strains, we crossed these strains to the piRNA sensor strain, and the experiment was performed as described above. When analyzing the tissue-specific rescue groups in this experiment, special care was taken to quantify only worms that carry the rescue array.

## Fluorescence microscopy

We used an *Olympus BX63* microscope for fluorescence microscopy assays. Experiments were filmed with a 10X objective lens, with an exposure time of 750 ms.

Since exposure to stress leads to developmental arrest and delays, worms in different conditions reached adulthood in different days. To avoid age bias between the different conditions, the developmental stages of all worms were tracked, and the worms under all conditions were imaged when they reached 'day one' adulthood.

## Small RNAs sequencing

### Collecting the worms

Total RNA was extracted from 'day one' adults. As stress induces variability in development even in isogenic worms' populations, tight synchronization of the worms' populations was achieved by picking L4 worms 1 day prior to collecting the worms for sequencing. Each experiment started by exposure of the generation before stress to an anti-*gfp* RNAi trigger to enable the phenotypic detection of resetting in the next generations. All sequencing experiments were done in triplicates (independent biological replicates).

### Libraries preparation

Worms were lysed using the TRIzol reagent (Life Technologies) followed by repetitive freezing, thawing, and vortex. The total RNA samples were treated with tobacco acid pyrophosphatase (TAP, Epicenter), to ensure 5' monophosphate-independent capturing of small RNAs. Libraries were prepared using the NEBNext Small RNA Library Prep Set for Illumina according to the manufacturer's protocol. The resulting cDNAs were separated on a 4% agarose E-Gel (Invitrogen, Life Technologies), and the 140–160 nt length species were selected. cDNA was purified using the MinElute Gel Extraction kit (QIAGEN). Libraries were sequenced using an Illumina NextSeq500 instrument.

## Quantification and statistical analysis

Fluorescence analysis; GFP Using the ImageJ *Fiji* 'measure' function (*Schindelin et al., 2012*), we measured the integrated density of the three germline nuclei closest to spermatheca in each worm,

as well as a mean background measurement of the worm in the germline's vicinity. If less than three germline nuclei were visible, a measurement was taken in the estimated location of the germline instead. The corrected total cell fluorescence (CTCF) of each germline nucleus was calculated as previously described (*Hammond, 2014*). We used the mean of all three CTCF scores as the worm's fluorescence score.

### Fluorescence analysis; piRNAs- and endo-siRNAs-derived silencing

Using *Fiji* (*Schindelin et al., 2012*), we have measured the integrated density of the whole worm, as well as one background measurement per worm. The corrected total fluorescence (CTF) of each worm was calculated as described above.

### Statistical analyses

Due to the highly variable nature of small RNA inheritance and erasure, we have elected to analyze our data using the nonparametric two-tailed Dunn's multiple comparison test. We corrected for multiple comparisons using the Benjamini-Hochberg False Discovery Rate (FDR), with an FDR of 0.05. The q-values reported in this study are adjusted to multiple comparisons. In the figures, not significant (ns) indicates $q \geq 0.05$, (*) indicates $q < 0.05$, (**) indicate $q < 0.01$, and (***) indicate $q < 0.001$. Unless marked otherwise in the figures, comparisons were made between each stressed group and its corresponding unstressed control group.

In the tissue-specific rescue experiments (*Figure 5D*), where a particularly large number of worms was measured (N = 3193), we encountered increased sensitivity during hypothesis testing and the detection of statistically significant effects that may not be biologically significant. We therefore chose to calculate and report effect size measures for each comparison in addition to corrected p-values.

The Cliff's Delta non-parametric effect size measure was used to estimate effect sizes for each comparison (*Cliff, 1993*; *Kromrey and Hogarty, 1998*). Cliff's Delta ranges between $-1$ and 1, where the sign indicates the direction of the effect, and the magnitude indicates the effect size. An absolute value of 1 indicates that there is no overlap between the two groups, whereas a value of 0 indicates that the groups' distributions overlap completely.

### Box plot graphs

Data are the individual worms' mean germline CTCF, with each dot representing the measurement of a single worm. Data are represented as medians, with box limits representing the 25th (Q1) and 75th (Q3) percentiles, notch representing a 95% confidence interval, and whiskers indicating Q1-1.5*IQR and Q3+1.5IQR. The minimal value of each experiment was set at 0 and the other values were adjusted accordingly.

### Small RNA-seq analysis

The Illumina fastq output files were first assessed for quality, using FastQC (*Andrews, 2010*), and compared to the FastQC-provided example of small RNA sequencing results. The files were then assigned to adapters clipping using Cutadapt (*Martin, 2011*) and the following specifications were used: cutadapt -m 15 -a AGATCGGAAGAGCACACGTCT input.fastq > output.fastq -m 15 discard reads which are shorter than 15 nucleotides after the adapter clipping process -a AGATCGGAAGAGCACACGTCT the 3' adapter sequence used as a query. The clipped reads were then aligned against the *ce11* version of the *C. elegans* genome using ShortStack (*Shahid and Axtell, 2014*) using the default settings:

$$ShortStack --readfile\ Input.fastq --genomefile\ Ce11Reference.fa$$

Next, we counted reads which align in the sense or antisense orientation to genes. Since stress is known to affect the abundance of structural RNA molecules, we omitted reads which align to structural genes from our analyses. We used the python-based script HTSeq-count (*Anders et al., 2014*) and the Ensembl-provided gff file (release-95), using the following command:

Antisense HTSeq.scripts.count `-stranded`=reverse –mode=union input.sam GENES.gff > output.txt

Sense HTSeq.scripts.count `-stranded`=yes –mode=union input.sam GENES.gff > output.txt

We then assigned the summarized counts for differential expression analysis using the R package DESeq2 (*Love et al., 2014*) and limited the hits for genes that were shown to have an FDR < 0.1. Normalization of the total number of reads in each sample and the total number of reads which align to the different types of genomic features (*Figure 5B*) was generated based on the SizeFactor normalization provided by the DESeq2 package (the median ratio method).

## mRNA-seq analysis

The Illumina fastq files were downloaded from GEO using *fastq-dump*. The files were then assigned to quality trimming using CutAdapt version 2.5 (*Martin, 2011*) and the following specifications were used: cutadapt -m 15 –q 30 –trim-n input.fastq > output.fastq -**m 15** discard reads which are shorter than 15 nucleotides after the adapter clipping process -**q 30** trim bases with a quality score below 30 from the 3' end –trim-n trim flanking 'N' bases from each read.

In the osmotic stress data and the anti-*skn-1* RNAi data, the reads still contained single-end adaptors and therefore the following specifications were added to the CutAdapt call: -a TruSeqRead1=AGATCGGAAGAGCACACGTCTGAACTCCAGTCA the 3' adapter sequence used as query.

In the heat shock data, the reads still contained paired-end adaptors and therefore the following specifications were added to the CutAdapt call: -a TruSeqRead1=AGATCGGAAGAGCACACGTCTGAACTCCAGTCA the queried adapter sequence for read #1 A TruSeqRead2=AGATCGGAAGAGCGTCGTGTAGGGAAAGAGTG the queried adapter sequence for read #2.

The clipped reads were then aligned against the *ce11* version of the *C. elegans* genome using HISAT2 version 2.1.0 (*Kim et al., 2019*) using the default settings, in single-end mode for single-end datasets and paired-end more for paired-end datasets.

The SAM files generated by HISAT2 were then sorted by name and converted into BAM files using samtools version 1.7, with the command: samtools sort -o ${filename}_sorted.bam -n ${fileename}.sam'.

Next, we counted reads which align in the sense or antisense orientation to genes. We used the python-based script HTSeq-count version 0.11.1 (*Anders et al., 2014*) and the Ensembl-provided gff file (release-95), using the following commands:

Unstranded datasets htseq-count –stranded=no –mode=intersection-nonempty –secondary-alignments=ignore sorted_input.bam GENES.gff > output.txt
Stranded datasets htseq-count –stranded=yes –mode=intersection-nonempty –secondary-alignments=ignore sorted_input.bam GENES.gff > output.txt

The heat shock data, which was supplied in the reverse orientation, was counted using the command:

htseq-count –stranded=reverse –mode=intersection-nonempty –secondary-alignments=ignore sorted_input.bam GENES.gff > output.txt

We then assigned the summarized counts for differential expression analysis using the R package DESeq2 version 1.24.0 (*Love et al., 2014*) and limited the hits for genes that were shown to have an FDR < 0.1.

Downstream filtering and enrichment analyses were performed using the python package RNAlysis version 1.3.5 (*Teichman, 2019*).

## Plotting H3K9me2 and H3K9me3 profiles

Chip-Seq data were downloaded from GSE87522. The transcription start site (TSS) of all the protein-coding genes in *C. elegans* were extracted from UCSC Genome Browser. We aligned all the genes according to their Transcription Start Sites (TSS) and extracted the signal along the flanking regions of 500 base pairs upstream and downstream of the TSS. In the case of genes with two transcripts or more, we averaged the histone modification signal of all the corresponding transcripts. The signal of individual genes was determined as the averaged signal of the two published replicates of each chromatin modification.

## Acknowledgements

We thank all members of the Rechavi lab for fruitful discussions and their support. Some strains were provided by the CGC, which is funded by NIH Office of Research Infrastructure Programs (P40 OD010440). We thank the Richard Roy lab for providing strain MR1175 (*aak-1/2*) and the Julie Ahringer lab for providing strain JA1527. We thank Yoav Ze'evi (statistics unit, Yoav Benjamini's group) for his help with the statistical analysis. OR is thankful to the Adelis Foundation grant #0604916191. GT is grateful to the Milner Foundation, and LH-Z is thankful to the Clore Foundation. The Rechavi lab is funded by ERC grant #335624 and the Israel Science Foundation (grant#1339/17).

## Additional information

### Funding

| Funder | Grant reference number | Author |
| --- | --- | --- |
| Clore Foundation | Graduate Student Fellowship | Leah Houri-Zeevi |
| Milner Foundation | Graduate Student Fellowship | Guy Teichman |
| European Research Council | #335624 | Oded Rechavi |
| Israel Science Foundation | #1339/17 | Oded Rechavi |
| Adelis Foundation | #0604916191 | Oded Rechavi |

The funders had no role in study design, data collection and interpretation, or the decision to submit the work for publication.

### Author contributions

Leah Houri-Zeevi, Conceptualization, Data curation, Formal analysis, Investigation, Visualization, Writing - original draft, Project administration, Writing - review and editing; Guy Teichman, Conceptualization, Data curation, Formal analysis, Investigation, Visualization, Writing - original draft, Writing - review and editing; Hila Gingold, Software, Visualization; Oded Rechavi, Conceptualization, Supervision, Funding acquisition, Writing - original draft, Writing - review and editing

### Author ORCIDs

Leah Houri-Zeevi (iD) https://orcid.org/0000-0003-2903-5082
Guy Teichman (iD) https://orcid.org/0000-0003-2285-1343
Oded Rechavi (iD) https://orcid.org/0000-0001-6172-3024

### Decision letter and Author response

Decision letter https://doi.org/10.7554/eLife.65797.sa1
Author response https://doi.org/10.7554/eLife.65797.sa2

## Additional files

### Supplementary files

• Supplementary file 1. A total of 281 targets of small RNAs which were affected across all stress conditions *at the stress generation*. Table presents DESeq2 comparison of Control vs. Stress samples. Related to *Figure 4*.

• Supplementary file 2. Ten targets of small RNAs which were affected across all stress conditions *at the next generation*. Table presents DESeq2 comparison of Control vs. Stress samples. Related to *Figure 4*.

• Supplementary file 3. Enrichment table for the 281 targets of stress-affected genes, generated using WormExp (*Yang et al., 2016*). Related to *Figure 4*.

- Supplementary file 4. Seventy-three epigenetic-related genes significantly downregulated under all stress conditions.
- Supplementary file 5. Thirty-five epigenetic-related *skn-1*-dependent stress-dependent genes.
- Supplementary file 6. Enrichment table for *skn-1*-dependent stress-dependent genes, generated using RNAlysis (*Teichman, 2019*).
- Supplementary file 7. Sixty-three epigenetic-related genes whose putative promoter regions contain the binding motif sequence of SKN-1, generated using TF2DNA (*Pujato et al., 2014*).
- Transparent reporting form

### Data availability

Sequencing data have been deposited in GEO under accession codes GSE129988.

The following dataset was generated:

| Author(s) | Year | Dataset title | Dataset URL | Database and Identifier |
|---|---|---|---|---|
| Teichman G, Gingold H, Rechavi O, Zeevi HL | 2020 | Stress resets transgenerational small RNA inheritance | https://www.ncbi.nlm. nih.gov/geo/query/acc. cgi?acc=GSE129988 | NCBI Gene Expression Omnibus, GSE129988 |

The following previously published datasets were used:

| Author(s) | Year | Dataset title | Dataset URL | Database and Identifier |
|---|---|---|---|---|
| McMurchy AN, Stempor P, Gaarenstroom T, Wysolmerski B, Dong Y, Aussianikava D, Appert A, Huang N, Kolasinska-Zwierz P, Sapetschnig A, Miska EA, Ahringer J | 2017 | A team of heterochromatin factors collaborates with small RNA pathways to combat repetitive elements and germline stress [ChIP-seq] | https://www.ncbi.nlm. nih.gov/geo/query/acc. cgi?acc=GSE87522 | NCBI Gene Expression Omnibus, GSE87522 |
| Dodd W, Tang L, Lone JC, Wimberly K, Wu CW, Consalvo C, Wright JE, Pujol N, Choe KP | 2018 | Role of SKN-1 in dpy-7 and osmotic gene induction | https://www.ncbi.nlm. nih.gov/geo/query/acc. cgi?acc=GSE107704 | NCBI Gene Expression Omnibus, GSE107704 |
| Finger F, Ottens F, Springhorn A, Drexel T, Proksch L, Metz S, Cochella L, Hoppe T | 2019 | RNA-seq: WT under proteotoxic stress conditions | https://www.ncbi.nlm. nih.gov/geo/query/acc. cgi?acc=GSE124178 | NCBI Gene Expression Omnibus, GSE124178 |
| Schreiner WP, Pagliuso DC, Garrigues JM, Chen JS, Aalto AP, Pasquinelli AE | 2019 | RNA Sequencing of CTRL and Heat Stressed C. elegans [RNA-seq] | https://www.ncbi.nlm. nih.gov/geo/query/acc. cgi?acc=GSE132838 | NCBI Gene Expression Omnibus, GSE132838 |
| Steinbaugh MJ, Narasimhan SD, Robida-Stubbs S, Moronetti Mazzeo LE, Dreyfuss JM, Hourihan JM, Raghavan P, Operaña TN, Esmaillie R, Blackwell TK | 2015 | RNA-seq analysis of germline stem cell removal and loss of SKN-1 in C. elegans | https://www.ncbi.nlm. nih.gov/geo/query/acc. cgi?acc=GSE63075 | NCBI Gene Expression Omnibus, GSE63075 |

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

# Appendix 1

## Appendix 1—key resources table

| Reagent type (species) or resource | Designation | Source or reference | Identifiers | Additional information |
|---|---|---|---|---|
| Chemical compound, drug | Levamisole hydrochloride | Sigma | L0380000 | |
| Chemical compound, drug | Trizol Reagent | Life Technologies | 15596026 | |
| Chemical compound, drug | Phenol Chloroform Isoamyl | Sigma | P2069 | |
| Other | Heavy Phase Lock tube | QuantaBio | 23028330 | |
| Chemical compound, drug | Ultra Pure Glycogen | ThermoFisher | 10814010 | |
| Peptide, recombinant protein | RNA 5' Polyphosphatase | Epicenter | RP8092H | |
| Commercial assay, kit | NEBNext Multiplex Small RNA Library Prep Set for Illumina | New England Biolabs | E7300 | |
| Commercial assay, kit | TapeStation screen tapes | Agilent | 5067–5582<br>5067–5588 | |
| Commercial assay, kit | TapeStation reagents | Agilent | 5067–5583<br>5067–5589 | |
| Chemical compound, drug | E-Gel 4% agarose | Life Technologies | G401004 | |
| Commercial assay, kit | MinElute DNA purification kit | Qiagen | 28006 | |
| Chemical compound, drug | RNase free Nuclease-free water | Ambion | AM9932 | |
| Peptide, recombinant protein | NlaIII | New Englang BioLabs (NEB) | 631207 | |
| Commercial assay, kit | TG NextSeq 500/550 High Output Kit v2 (75 cycles) | Illumina | TG-160–2005 | |
| Strain, strain background (*Caenorhabditis elegans*) | *C. elegans*: Strain SX1263: unc-119(ed3) III; mex-5::gfp::h2b::tbb-2 II | The Eric Miska lab (*Sapetschnig et al., 2015*) | SX1263 | |
| Strain, strain background (*C. elegans*) | *C. elegans*: Strain EG6089: unc-119(ed3) III; oxTi38[cb-unc-119(+) Ppie-1::GFP] IV | The Eric Miska lab (*Sapetschnig et al., 2015*) | EG6089 | |
| Strain, strain background (*C. elegans*) | *C. elegans*: Strain JA1527: weSi14 [*Pmex-5::mCherry::(Gly)5Ala/his-58/tbb-2* 3'UTR; cb-*unc-119*(+)] IV | The Julie Ahringer lab (*Zeiser et al., 2011*) | JA1527 | |
| Strain, strain background (*C. elegans*) | *C. elegans*: Strain GR1720: mgSi4 [(pCMP2) *ubl-1p::GFP::siR-1-sensor-ubl-1–*3'UTR + Cbr-*unc-119*(+)] IV | Caenorhabditis Genetics Center | WB strain: GR1720 | |
| Strain, strain background (*C. elegans*) | *C. elegans*: Strain KB3: *kgb-1* (um3) IV | Caenorhabditis Genetics Center | KB3 | |

*Continued on next page*

*Appendix 1—key resources table continued*

| Reagent type (species) or resource | Designation | Source or reference | Identifiers | Additional information |
|---|---|---|---|---|
| Strain, strain background (*C. elegans*) | *C. elegans*: Strain KB3: *kgb-1* (um3) IV crossed with Strain SX1263 | This study | | |
| Strain, strain background (*C. elegans*) | *C. elegans*: Strain FK171: *mek-1* (ks54); *sek-1*(qd127) | Caenorhabditis Genetics Center | FK171 | |
| Strain, strain background (*C. elegans*) | *C. elegans*: Strain FK171: *mek-1* (ks54); *sek-1*(qd127) X crossed with Strain SX1263 | This study | | |
| Strain, strain background (*C. elegans*) | *C. elegans*: Strain CF1038: *daf-16*(mu86) I | Caenorhabditis Genetics Center | CF1038 | |
| Strain, strain background (*C. elegans*) | *C. elegans*: Strain CF1038: *daf-16*(mu86) I, crossed with Strain SX1263 | This study | | |
| Strain, strain background (*C. elegans*) | *C. elegans*: Strain KU4: *sek-1* (km4) X | Caenorhabditis Genetics Center | KU4 | |
| Strain, strain background (*C. elegans*) | *C. elegans*: Strain KU4: *sek-1* (km4) X, crossed with Strain SX1263 | This study | | |
| Strain, strain background (*C. elegans*) | *C. elegans*: Strain KU25: *pmk-1* (km25) IV | Caenorhabditis Genetics Center | KU25 | |
| Strain, strain background (*C. elegans*) | *C. elegans*: Strain KU25: *pmk-1* (km25) IV, crossed with Strain SX1263 | This study | | |
| Strain, strain background (*C. elegans*) | *C. elegans*: Strain MR1175: *aak-1*(tm1944) III, *aak-2*(ok524) X | The Richard Roy lab (*Demoinet et al., 2017*) | MR1175 | |
| Strain, strain background (*C. elegans*) | *C. elegans*: Strain MR1175: *aak-1*(tm1944) III, *aak-2*(ok524) X, crossed with Strain SX1263 | This study | | |
| Strain, strain background (*C. elegans*) | *C. elegans*: Strain PS355: *hsf-1* (sy441) I | Caenorhabditis Genetics Center | PS355 | |
| Strain, strain background (*C. elegans*) | *C. elegans*: Strain PS355: *hsf-1* (sy441) I, crossed with Strain SX1263 | This study | | |
| Strain, strain background (*C. elegans*) | *C. elegans*: Strain CB1370: *daf-2* (e1370) III | Caenorhabditis Genetics Center | CB1370 | |
| Strain, strain background (*C. elegans*) | *C. elegans*: Strain CB1370: *daf-2* (e1370) III, crossed with Strain EG6089 | This study | | |
| Strain, strain background (*C. elegans*) | *C. elegans*: Strain QV225: *skn-1* (zj15) IV | Caenorhabditis Genetics Center | QV225 | |
| Strain, strain background (*C. elegans*) | *C. elegans*: Strain QV225: *skn-1* (zj15) IV, crossed with Strain SX1263 | This study | | |
| Strain, strain background (*C. elegans*) | *C. elegans*: Strain MT13293: *met-2*(n4256) III | Caenorhabditis Genetics Center | MT13293 | |

*Continued on next page*

*Appendix 1—key resources table continued*

| Reagent type (species) or resource | Designation | Source or reference | Identifiers | Additional information |
|---|---|---|---|---|
| Strain, strain background (*C. elegans*) | *C. elegans*: Strain MT13293: *met-2*(n4256) III, crossed with Strain SX1263 | This study | | |
| Strain, strain background (*C. elegans*) | *C. elegans*: Strain RB1789: *met-2*(ok2307) III | Caenorhabditis Genetics Center | RB1789 | |
| Strain, strain background (*C. elegans*) | *C. elegans*: Strain RB1789: *met-2*(ok2307) III, crossed with Strain SX1263 | This study | | |
| Strain, strain background (*C. elegans*) | *C. elegans*: Strain KU4: *sek-1*(km4) X, crossed with strain JA1527 | This study | | |
| Strain, strain background (*C. elegans*) | *C. elegans*: Strain ZD202: *sek-1*(km4) X; qdEx8[*unc-119p::sek-1*(cDNA)::GFP::*unc-54*–3' UTR + *myo-2p::mStrawberry::unc-54*–3' UTR], crossed with strain JA1527 | This study | | |
| Strain, strain background (*C. elegans*) | *C. elegans*: Strain ZD193: *sek-1*(km4) X; qdEx4 [*ges-1p::sek-1*(cDNA)::GFP::*unc-54*–3' UTR + *myo-2p::mStrawberry::unc-54*–3' UTR], crossed with strain JA1527 | This study | | |
| Strain, strain background (*C. elegans*) | *C. elegans*: Strain OH14221: *met-2*(ot861[*met-2*::mKate2]) III | Caenorhabditis Genetics Center (*Patel and Hobert, 2017*) | OH14221 | |
| Strain, strain background (*C. elegans*) | *C. elegans*: Strain OH14221: *met-2*(ot861[*met-2*::mKate2]) III crossed with strain QV225 | This study | | |
| Sequence-based reagent | PCR 1-FWD: *mek-1/sek-1* | IDT | | TTTCCATCAACTCAGTCGCCG |
| Sequence-based reagent | PCR 1-REV1: *mek-1/sek-1* | IDT | | TTCATTAGTCAATTGGGTCAG |
| Sequence-based reagent | PCR 1-REV2: *mek-1/sek-1* | IDT | | CACTTTTCAATTAAGGTACAAC |
| Sequence-based reagent | PCR 2-FWD: *kgb-1* | IDT | | CCCTACTTTATAATGAGATGC |
| Sequence-based reagent | PCR 2-REV1: *kgb-1* | IDT | | TTCATTAGTCAATTGGGTCAG |
| Sequence-based reagent | PCR 2-REV2: *kgb-1* | IDT | | CACTTTTCAATTAAGGTACAAC |
| Sequence-based reagent | PCR 3-FWD: *daf-16* | IDT | | GTTCAGTAGACGGTGACCATCT |
| Sequence-based reagent | PCR 3-REV1: *daf-16* | IDT | | GCTTCGGCTTGAAAGATCAGTG |
| Sequence-based reagent | PCR 3-REV2: *daf-16* | IDT | | GTACGCCGTGGTCCGACTA |
| Sequence-based reagent | PCR 4-FWD: *skn-1* | IDT | | GAAGAGAATGCTCGATATGAAG |
| Sequence-based reagent | PCR 4-REV: *skn-1* | IDT | | TTTCAGTCGTTTATAAGAGAGC |
| Sequence-based reagent | PCR 5-FWD: *aak-1* | IDT | | ATCGATACGGAACCAACTG |

*Continued on next page*

*Appendix 1—key resources table continued*

| Reagent type (species) or resource | Designation | Source or reference | Identifiers | Additional information |
|---|---|---|---|---|
| Sequence-based reagent | PCR 5-REV: *aak-1* | IDT | | GGGTATGGTA GTACCAATAGG |
| Sequence-based reagent | PCR 6-FWD: *aak-2* | IDT | | CGATAGCAC AGACAACAGTTCG |
| Sequence-based reagent | PCR 6-REV: *aak-2* | IDT | | GATGGTGGC CCTCTTCATC |
| Sequence-based reagent | PCR 7-FWD1: *daf-18* | IDT | | AGGGTAATGCA TTTCAGCAC |
| Sequence-based reagent | PCR 7-FWD2: *daf-18* | IDT | | CCCGCATATA AACTGGAAATGTG |
| Sequence-based reagent | PCR 7-REV: *daf-18* | IDT | | CAAATACGT CAGTTTCAACGTG |
| Sequence-based reagent | PCR 8-FWD: *pmk-1* | IDT | | CTATAAGTTGC CATGACCTC |
| Sequence-based reagent | PCR 8-REV: *pmk-1* | IDT | | GCTCCCATCA ACATTGATAC |
| Sequence-based reagent | PCR 9-FWD1: *sek-1* | IDT | | CTAGAATAAG TGCTATGCTAG |
| Sequence-based reagent | PCR 9-FWD2: *sek-1* | IDT | | GTTGTCTAAG TATAATTGTCC |
| Sequence-based reagent | PCR 9-REV: *sek-1* | IDT | | TGATTGATTAT AACTACGAGG |
| Sequence-based reagent | PCR 10-FWD1: *met-2* | IDT | | TTTACTGTC ACATCACCTGC |
| Sequence-based reagent | PCR 10-FWD2: *met-2* | IDT | | AAGCAGATGT TTGTCAGAATCC |
| Sequence-based reagent | PCR 10-REV: *met-2* | IDT | | AGCAGCATTC ATCTTCGC |
| Software, algorithm | FastQC | *Andrews, 2010* | | |
| Software, algorithm | Cutadapt | *Martin, 2011* | | |
| Software, algorithm | Shortstack | *Shahid and Axtell, 2014* | | |
| Software, algorithm | HTSeq count | *Anders et al., 2014* | | |
| Software, algorithm | R Deseq2 | *Love et al., 2014* | | |
| Software, algorithm | RNAlysis | *Teichman, 2019* | | Version 1.3.5 |
| Software, algorithm | Fiji | *Schindelin et al., 2012* | | |
| Software, algorithm | MATLAB R2018b | MATLAB | | Version R2018b |
| Software, algorithm | IoSR-Surry MatlabToolbox (BoxPlot function) | Institute of Sound Recording, University of Surrey | https://github.com/ IoSR-Surrey/ MatlabToolbox | Version 2.8 |
| Software, algorithm | GraphPad Prism 8 | GraphPad Software | http://www.graphpad.com/ | Version 8.0.0 |

