## [Decision Letter]

Our editorial process produces two outputs: (i) public reviews designed to be posted alongside the preprint for the benefit of readers; (ii) feedback on the manuscript for the authors, including requests for revisions, shown below.

[Editors’ note: the authors submitted for reconsideration following the decision after peer review. What follows is the decision letter after the first round of review.]

Thank you for submitting your work entitled "Stress Resets Transgenerational Small RNA Inheritance" for consideration by *eLife*. Your article has been reviewed by three peer reviewers, and the evaluation has been overseen by a Reviewing Editor and a Senior Editor. The reviewers have opted to remain anonymous.

Our decision has been reached after consultation between the reviewers. Based on these discussions and the individual reviews below, we regret to inform you that your work will not be considered further for publication in *eLife*.

The individual reviews are appended below and I am also summarizing the main issues that all reviewers agreed would be the minimum set of experiments/rewriting that would be necessary for the manuscript to be reconsidered as a new submission.

1) Expand on the connection among MET-2/MAPK/SNK-1 and RNAi pathways.

2) Establish tissue specificity (gut? neurons?)

3) Expand on the insulin signaling pathway – perform the transgene expression assay in daf-2; daf-16 and daf-2; skn-1 double mutants

4) Show as controls in the P0 generation what the effect of the mapk, skn-1 and met-2 mutations is on the GFP reporter strain

5) Establish whether the phenomenon is terminated in the F4 or F5 generation

6) Writing revision to make the paper more cohesive and integrate the findings conceptually with published literature (including your 2016 Cell paper), and explain how the different regulators identified are connected to each other. All reviewers found the paper to be somewhat disjointed.

7) Expand on how different stressors may affect small RNA pools in different ways.

Reviewer #1:

Here, Houri-Ze'evi, et al. treated progeny of parents that had inherited small RNA response (silencing of an artificial, single-copy GL-expressed gfp with anti-gfp dsRNA) with 3 stresses – heat shock, hyperosmolarity, or starvation – starting at L1, and examined gfp silencing. All three treatments reduced silencing (visible as increased GFP fluorescence) in subsequent generations (F1-F3).

The authors tested resetting of endogenous (endo-siRNA) and piRNAs using an endo-siRNA sensor target sequence and piRNA recognition sites, respectively. Again, all 3 stressors reset both in the same generation, but did not reset the effect transgenerationally, suggesting that exogenous RNAi resetting functions through a different mechanism than endogenous.

Next, they tested adults, which also led to resetting. However, only the F1 generation, not F2, is susceptible to resetting (how? Why?), revealing a critical period for resetting susceptibility.

Reversal of the stress with RNAi treatment does not result in resetting, nor does simply changing conditions.

The authors then went on to examine mutants that might be defective in stress responses or in resetting; MAPK genes and skn-1 are required for resetting.

Small RNA-seq from stressed worms and their progeny showed a decrease overall with stresses, and reveals some potential classes of genes, including targets of the mutator genes, and overlap with classic stress response pathways (dauer, IIS).

Overall, this work presents some interesting phenomena and moves towards explaining how it might work through the identification of a critical period and some genes that are required. However, several large gaps remain to be addressed.

The major gaps that need to be addressed are in the MAPK/SKN-1 function (how? Connections? Where?) and in general, the route of communication of these resetting process, particularly in light of the lab's most recent findings in neurons.

1) It seems that the ultimate conclusions regarding MAPK/SKN-1 were left hanging. Does MAPK pathway regulate SKN-1? (nuclear localization?) How do MAPK and SKN-1 function to reset? And where?

2) It would be good to compare transcription in the skn-1 mutants than can't reset with wt under the same resetting conditions, to identify possible mechanisms by which resetting occurs.

3) What is the connection between MAPK, SKN-1, and mutator genes? Does SKN-1 regulate these mutator genes directly? Is there a functional connection with MET-2?

4) Perhaps most critically, where (in which tissues) are these effects taking place? Does the stress response happen in somatic tissues (and which ones? intestine? neurons?) and communicate to the germline? The authors cite papers that discuss gene functions in the nervous system (e.g., Bishop and Guarente) but have not shown that these genes function in the nervous system to reset. Tissue-specific rescues of the required genes should be done to address this question.

5) Do the answers to these questions explain why there are differences between the exogenous and endogenous siRNA and piRNA pathways? or critical period/generation? This aspect was also left unresolved.

Reviewer #2:

Transgenerational inheritance of gene expression states has been demonstrated in plants and animals. In *C. elegans*, transgenerational inheritance often lasts 3-5 generations, and increasing evidence suggests that the inherited gene expression states are mediated by RNA interference pathways. How this inheritance is maintained and "reset" after approximately three generations has been an open question in the small RNA field. Understanding how environmental stress can propagate gene expression states across generations has become an important field of study with regard to human disease, particularly for diseases with fetal origins such as metabolic programming disorders. In a previous publication, this lab showed that the termination of transgenerational gene expression states is a regulated process. In this follow-up manuscript, the authors show that environmental stresses- starvation, high osmolarity, and high temperature- can disrupt the inheritance of gene expression through changes in small RNA levels. The authors also demonstrate that the stress-induced disruption of inherited gene expression is dependent upon the MAPK pathway and the MET-2 histone H3K9 methyltransferase.

While I think this manuscript describes an important finding with regards to small RNA inheritance, I think the results need to be more conceptually integrated with previous work. For example, numerous studies have examined transgenerational inheritance after heat stress and the role of MET-2 in establishing H3K9me2 marks in the early embryo and in the adult germ line. How does this study complement or disagree with previous findings? The answer to this question might help in developing a more testable model of how MAPK and MET-2 function are mechanistically linked to small RNA production. Related to this comment, the authors do not address how this work relates to their previous paper (Houri-Ze'evi et al., 2016), which showed that dsRNA-induced RNAi triggered the RNAi-mediated downregulated of RNAi genes. Does stress in L1 progeny of adults treated with dsRNA further down-regulate these RNAi genes, allowing the "reset" of gene expression to happen earlier? In addition, I found the term "reset" to be somewhat of a misnomer, which led to some confusion while reading the manuscript. Dr. Rechavi demonstrated as a postdoc that L1 starvation leads to transgenerational inheritance of a small RNA response to stress. Thus, it seems that stress does not exactly "reset" the small RNA populations to a "normal" state, but rather the worms re-prioritize their small RNA repertoire for a greater need, i.e. starvation/osmolarity/temperature stress trumps silencing a single transgene.

The final major comment about the manuscript is that the authors have not fully explored the potential contribution of the insulin signaling pathway to this process, despite reporting an effect on small RNA "resetting" in the daf-2 mutant and a significant overlap of insulin regulated genes with their identified 281 stress-affected genes. Insulin signaling regulates SKN-1 function, which could potentially explain how the authors observe a generalized small RNA response to different stresses (this review might be helpful: doi: 10.1016/j.freeradbiomed.2015.06.008). The authors should perform their transgene expression assay in daf-2;daf-16 and daf-2;skn-1 double mutants to examine the potential role of insulin signaling and DAF-16 in this process. It would also be useful to examine the overlap of their 281 stress-responsive genes with DAF-16 class I and class II genes.

Reviewer #3:

In the paper entitled "Stress Resets Transgenerational Small RNA Inheritance" Houri-Ze'evi L, Teichman G et al. examine the interaction between multiple heritable phenotypes by knocking down a heritable GFP reporter and examining its interaction with other stresses, such as starvation and high temperature, which cause transgenerationally heritable phenotypes. They demonstrate that exposing worms to stresses inhibits the transgenerational silencing of the GFP reporter strain they use. They further demonstrate that deletion of genes involved in the MAPK pathway, the skn-1 transcription factor and the putative H3K9 methyltranferase met-2 eliminate the differential response in the F1 and F2 generations after exposure to stress and the GFP reporter silencing. They also sequence the small RNAs in the P0 and F1 generation with and without the added stresses. This is an extremely exciting finding but I feel that it should be a little more flushed out before publication. How are these different pathways connected? The system itself should also be a little more rigorously probed. Some detailed suggestions to help ameliorate the paper are listed below.

1) For Figure 1 How long does the phenotype persist (what happens in F4 and F5)? If worms are persistently stressed (across many generations in advance of the experiment) does this blunt the effect of the GFP knockdown in the first generation or its transmission? Basically is the reverse true as well?

2) For Figure 3 it suggests to me that there is only so much bandwidth that the heritable silencing can occupy so if it is activated to respond to a dramatic stress than it isn't going to be as efficient at having heritable silencing of some GFP transgene. It seems like testing the magnitude of the different stresses would be good to help tease this apart. Is there a certain amount of heat shock or starvation which doesn't elicit a heritable response? Can you tune that down to such an extent that there is no effect on the heritable GFP knock-down? Also I couldn't really figure it out from the text if this was done with one GFP reporter strain or multiple ones? If it was done with multiple strains maybe explain that a little more thoroughly, if it was only done with one strain it would strengthen the paper to do it with another one.

3) For Figure 4 it would be nice to delve a little deeper. Does something happen to the MAPK protein levels in response to stress in one generation? It seems as though there isn't much connection between the different mediators of the stress response. Can the authors connect them more either experimentally or at least by describing in more detail their theories.

4) Again in Figure 5 when discussing different stressors affecting small RNA pools in different ways, it's unclear as to whether this is due to a different magnitude of stress or different pathways. Can the authors delve deeper into this? What is different about the small RNAs? Maybe more description of what the small RNAs are rather than the numbers and types GO analysis of enrichment in each category and the statistics and figure should represent that info.

[Editors’ note: further revisions were suggested prior to acceptance, as described below.]

Thank you for submitting your article "Stress Resets Ancestral Heritable Small RNA Responses" for consideration by *eLife*. Your article has been reviewed by three peer reviewers, and the evaluation has been overseen by a Reviewing Editor and Kevin Struhl as the Senior Editor. The reviewers have opted to remain anonymous.

Reviewer #1 (Recommendations for the authors):

It was unclear whether the germline rescue of sek-2 was tested for its contribution to the observed effects.

Some but not all of the RNA-seq data are described in terms of enrichment and significance; it would be preferable to see this in all the discussions of transcriptional data.

Reviewer #2 (Recommendations for the authors):

1) The figures show that stress treatment of F1 progeny of P0 adults treated with dsRNA resulted in de-silencing of the GFP transgene, presumably through changes in exo-siRNA levels. In Figure 1C, the transgene de-silencing appears complete by the F4 generation, compared to control progeny that do not completely de-silence the transgene until F6 generation. [It would be helpful in these experiments, where appropriate, to perform an ANOVA to compare across generations.] In contrast, when endo-siRNA responses were examined using an endo-siRNA and piRNA transgene sensor (Figure 2), the de-silencing was only observed F1 generation that experienced the stress. These differences indicate two different mechanisms regulating siRNAs during and after stress. Exo-siRNAs appear to be “turned over” much faster due to stress conditions, whereas siRNAs were only affected during the stressed generation. These two mechanisms are lumped together in the manuscript as one mechanism, although the authors do somewhat make distinctions in parts of the Results. This creates problems later in Figure 5D, when rescue of sek-1 is examined using a piRNA sensor strain, while Figure 5C examines the role of mutants regulating exo-siRNAs. For consistency, the experiment in Figure 5D should use a transgene reporter regulated by exo-siRNAs as well. I also think it would be more helpful to the reader to discuss these mechanisms separately in the manuscript.

2) In Figure 4, the small RNA levels were examined in stressed adults and their non-stressed progeny. They show an overall decrease in sRNAs for all three stresses of stressed adults, but show variable changes for each stress in the non-stressed progeny. This data is largely consistent with the results of Figure 2. Since Figure 3 explored the dynamics of how sRNA populations could be altered by different types of stress, I think it would be informative in Figure 4 to also show the levels of exo-RNAs targeting GFP in these populations. The dynamics of how exo-RNAs and endo-siRNAs may compete for resources during a stress response could be explored more here. In addition, the authors mention the lack of reproducibility among the replicates in Figure 4C. The control lanes look very similar, yet replicates of same stress do not cluster together. Are the replicates for each stress collected at the same time clustering together instead? Perhaps another environmental influence was contributing to these sRNA differences.

3) The nomenclature of a subset of the siRNAs identified in Figure 4C as "stress-reset" I find confusing. I think of a "reset" as return to baseline, whereas in this case, the worms are exhibiting a decrease from baseline for those siRNAs that are maintained over generations to protect germ line integrity.

4) Figure 1 is blurry and pixelated, which may have occurred during the conversion process.

5) In Figure 1C, 2B, (and others)- it would be helpful to have GFP levels of the transgene with empty vector treatment to compare if populations were actually "reset" or if the populations were gradually de-silencing over generations.

6) In Figure 5, the mutant names do not line up with the boxes.

7) In Figure 5C- mek-1;sek-1 F2 looks like wildtype F6 in Figure 1. Perhaps this strain has a failure to inherit small RNAs in addition to failure to reset.

8) Introduction – nuclear small RNAs “promote modification of” chromatin

9) Subsection “Stress, and not any change in growth conditions, leads to the

resetting of ancestral small RNA silencing”/Figure 3: it would be helpful to indicate "mild stress" to distinguish it from the harsher stresses used in other figures for clarity

10) Are RNAi genes targeted by MET-2, resulting in maintained strong silencing of the transgene?

11) Results final paragraph – : should be changed to period to start new sentence

12) In the Discussion, it would be helpful to compare the results presented in this manuscript to some of the studies that have examined transgenerational phenotypes resulting from similar stresses.

Reviewer #3 (Recommendations for the authors):

The authors for the most part answered my questions and I think the paper is basically ready for publication. I would make a few minor changes as suggested below. Additionally I still feel that the portion with MET-2 is not sufficiently fleshed out nor linked to the rest of the manuscript. I would suggest removing this final figure as the connection to skn-1 etc is tenuous still.

– First paragraph in subsection “Stress-induced resetting of target-specific small RNAs” ends prematurely "rapamycin exposure and changes in…"

– Third paragraph of the same section miscited as Figure 5E should be 4E.

– Something messed up happened in the citations. There is a citation Chen Qi, Yan Menghong, etc. which seems to just include a long list of researchers not a specific paper. Similarly the citation Kishimoto S, Uno M,.… is a list of people not in the paper that is cited.

– If the authors insist on including the MET-2 portion of the manuscript (again which I feel is out of place and doesn't yet belong in this paper), when discussing MET-2 and citing Kerr et al., 2014 and work from your own lab it seems appropriate to also cite work from Susan Gasser, Susan Mango, Yang Shi, and Susan Strome, all of whose work came out either before or at the same time.

---

## [Author Response]

[Editors’ note: the authors resubmitted a revised version of the paper for consideration. What follows is the authors’ response to the first round of review.]

Reviewer #1:Here, Houri-Ze'evi, et al. treated progeny of parents that had inherited small RNA response (silencing of an artificial, single-copy GL-expressed gfp with anti-gfp dsRNA) with 3 stresses – heat shock, hyperosmolarity, or starvation – starting at L1, and examined gfp silencing. All three treatments reduced silencing (visible as increased GFP fluorescence) in subsequent generations (F1-F3).The authors tested resetting of endogenous (endo-siRNA) and piRNAs using an endo-siRNA sensor target sequence and piRNA recognition sites, respectively. Again, all 3 stressors reset both in the same generation, but did not reset the effect transgenerationally, suggesting that exogenous RNAi resetting functions through a different mechanism than endogenous.Next, they tested adults, which also led to resetting. However, only the F1 generation, not F2, is susceptible to resetting (how? Why?), revealing a critical period for resetting susceptibility.Reversal of the stress with RNAi treatment does not result in resetting, nor does simply changing conditions.The authors then went on to examine mutants that might be defective in stress responses or in resetting; MAPK genes and skn-1 are required for resetting.Small RNA-seq from stressed worms and their progeny showed a decrease overall with stresses, and reveals some potential classes of genes, including targets of the mutator genes, and overlap with classic stress response pathways (dauer, IIS).Overall, this work presents some interesting phenomena and moves towards explaining how it might work through the identification of a critical period and some genes that are required. However, several large gaps remain to be addressed.

We thank the reviewer for finding our work interesting, for the concise summary of our results, and for the constructive remarks that help improve the manuscript. We resolved all the reviewer’s suggestions and requests with new experiments and extensive analyses. We also made major structural changes in the manuscript that should improve the flow and clarity of the revised manuscript, in particular in the sections that explore the regulation of stress-induced resetting of heritable responses.

The major gaps that need to be addressed are in the MAPK/SKN-1 function (how? Connections? Where?) and in general, the route of communication of these resetting process, particularly in light of the lab's most recent findings in neurons.1) It seems that the ultimate conclusions regarding MAPK/SKN-1 were left hanging. Does MAPK pathway regulate SKN-1? (nuclear localization?) How do MAPK and SKN-1 function to reset? And where?

We made great efforts in the revision process to address this remark. The p38 MAPK pathway was previously shown to regulate SKN-1 through MAPK-dependent phosphorylation, which leads to nuclear localization of the transcription factor SKN-1 (Inoue et al., 2005a). We elaborate on this in the revised manuscript.

To investigate the reviewer’s question, we ran multiple experiments and analyses:

1) To assess whether stress-induced resetting of small RNAs depends on a MAPK response in a particular tissue, we used worms that express the MAPK protein SEK-1, which we showed is crucial for stress-induced resetting of small RNAs, specifically in neurons or in the intestine (using tissue-specific promoters). We found that a neuronal rescue of *sek-1*, but not an intestinal rescue, was sufficient for worms to reset small RNAs in response to stress. While this shows that SEK-1 can affect resetting non-cell autonomously (via neurons), we note that the general levels of silencing in worms which carry an intestinal rescue of sek-1 were reduced even in the absence of stress and thus we cannot exclude that stress-induced resetting in these worms is masked by generally low levels of silencing. These results are congruent with our lab’s recent findings about neuronal small RNAs (in which we showed the neurons can affect small RNA inheritance), which the reviewer mentioned (Posner et al., 2019). We elaborate on those results further in the revised manuscript (see Figure 5).

2) To discover the downstream effectors of SKN-1 and the MAPK pathway in the stress-induced resetting response, we analyzed mRNA sequencing data from wild-type and *skn-1* knockdown worms that were exposed to either control conditions or hyperosmotic stress (Dodd et al., 2018; Steinbaugh et al., 2015). We found that exposure to stress in general represses the expression of a considerable group of epigenetic and small RNA factors. We elaborate on those results further in the revised manuscript. More details regarding the specific factors can be found below in this letter (in response to comment #2 of Reviewer #1).

Furthermore, we found that while *skn-1* knockdown, on its own, did not affect the expression of any notable epigenetics-related genes, the stress-induced repression of multiple small RNA factors depends upon *skn-1*. We identified a group of small RNA factors (N=35) that are repressed in response to stress in a *skn-1*-dependent manner. These include, in particular, the argonaute protein *nrde-3,* and the epigenetic factors *rrf-3, rde-8* and *nyn-2*. We elaborate on those results further in the revised manuscript.

3) To identify other potential downstream effectors of *skn-1* that function in resetting, we examined the putative promoter regions of known small RNA factors for the established binding motif sequence of SKN-1 (Pujato et al., 2014). We found that the promoters of multiple (N=63) small RNA factors contain SKN-1 binding sites (including *met-2*). We elaborate on those results further in the revised manuscript. SKN-1 regulation might work directly or indirectly, affecting MET-2 via any of its many target genes. However, further work is required to fully capture the regulatory role of SKN-1 over MET-2 and other small RNA factors.

4) In addition, while we were revising this manuscript, additional evidence that SKN-1 functions in transgenerational inheritance was published, strengthening the link that we identified between SKN-1 and regulation of transgenerational epigenetic inheritance (Burton et al., 2020). SKN-1 functions were also found to be regulated transgenerationally by heritable small RNA (Ewe et al., 2020).

2) It would be good to compare transcription in the skn-1 mutants than can't reset with wt under the same resetting conditions, to identify possible mechanisms by which resetting occurs.

We thank the reviewer for this great suggestion. We analyzed mRNA sequencing data from wild-type and *skn-1* knockdown worms that were exposed to either control conditions or hyperosmotic stress (Dodd et al., 2018; Steinbaugh et al., 2015). We elaborated about the results of this analysis in response to the comment above, as well as in the revised manuscript. In summary, we found that a considerable group of small RNA factors that are significantly downregulated in response to stress in a *skn-1*-dependent manner. In particular, this group includes the argonaute protein *nrde-3* and the epigenetic factors *rrf-3, rde-8* and *nyn-2*.

3) What is the connection between MAPK, SKN-1, and mutator genes? Does SKN-1 regulate these mutator genes directly? Is there a functional connection with MET-2?

We thank the reviewer for these important questions. We took multiple approaches to answer these questions:

1) As we elaborated in response to Comment #1 of the reviewer, previous works have shown that the p38 MAPK pathway regulates the activity of SKN-1 through MAPK-dependent phosphorylations, which in turn leads to nuclear localization of SKN-1 (Inoue et al., 2005b).

2) As we elaborated in response to Comment #1 of the reviewer, we analyzed mRNA sequencing data from wild-type and *skn-1* knockdown worms which were exposed to either control conditions or hyperosmotic stress. We elaborate on those results further in the revised manuscript, and in the responses above.

Notably, we found that the genes *rde-8* and *nyn-2* are both repressed in response to stress in a *skn-1* dependent manner. The two proteins that these genes encode for were shown to localize to the Mutator foci and have a crucial role in the cleavage of mRNA targets and the recruitment of the RNA-dependent RNA polymerase complex (Tsai et al., 2015). These results indicate that *skn-1* has a role in regulating components of the Mutator foci.

3) As we elaborated in response to Comment #1 of the reviewer, we examined the putative promoter regions of epigenetic and small RNA factors for the SKN-1 binding motif (Pujato et al., 2014). We found that the promoter region of *met-2* contains multiple SKN-1 binding motifs. We elaborate on those results further in the revised manuscript.

4) We analyzed mRNA sequencing data from worms that were exposed to heat shock, hyperosmotic stress, or starvation (Dodd et al., 2018; Finger et al., 2019; Schreiner et al., 2019). Among other findings, we discovered that a notable group of epigenetic and small RNA factors are repressed in response to any of those stress conditions. In particular, we found that the Mutator genes *mut-2* and *mut-16*, as well as the Mutator-interacting factor *rde-8*, are significantly downregulated in response to stress. We elaborate on those results further in the revised manuscript.

4) Perhaps most critically, where (in which tissues) are these effects taking place? Does the stress response happen in somatic tissues (and which ones? intestine? neurons?) and communicate to the germline? The authors cite papers that discuss gene functions in the nervous system (e.g., Bishop and Guarente) but have not shown that these genes function in the nervous system to reset. Tissue-specific rescues of the required genes should be done to address this question.

We thank the reviewer for these excellent ideas. To address this point, we used worms that express *sek-1*, a MAPK gene that we found to be required for stress-induced resetting of small RNAs, either in neurons or in the intestine, as suggested by the reviewer. We elaborated on these results in response to Comment #1 of the reviewer, but in short, we found that only neuron-specific expression of *sek-1* was sufficient to rescue stress-induced resetting, while intestinal-specific expression was not. While this shows that SEK-1 can affect resetting non-cell autonomously (via neurons), we note that the general levels of silencing in worms which carry an intestinal rescue of sek-1 were reduced even in the absence of stress and thus we cannot exclude that stress-induced resetting in these worms is masked by generally low levels of silencing. We elaborate on those results further in the revised manuscript (see Figure 5).

5) Do the answers to these questions explain why there are differences between the exogenous and endogenous siRNA and piRNA pathways? or critical period/generation? This aspect was also left unresolved.

We thank the reviewer for these interesting questions. Regarding the differences between the exogenous and endogenous small RNA pathways – we explained our hypothesis for the difference between exo-siRNAs and endo-siRNAs/piRNA in the manuscript and chose to elaborate on it further: unlike exogenous primary small RNAs which cannot be re-synthesized in the progeny, primary endo-siRNAs and piRNAs are encoded in the genome and do not depend on exogenous sources for their existence (Ambros et al., 2003; Cecere et al., 2012; Duchaine et al., 2006; Gu et al., 2012; Lee et al., 2006; Lemmens and Tijsterman, 2011; Ruby et al., 2006). The re-establishment of endo-siRNAs and piRNAs-mediated silencing in the next generations after stress indicates that these small RNAs can be transcribed de novo at each generation, and thus can compensate for stress-induced erasure of parental small RNA molecules, suggesting a fundamental difference in the “memory programs” of exogenous and endogenous transgenerational small RNA responses. We hope our explanation is now clearer.

Regarding the critical period: this is a very good question. We do not know the exact mechanism that explains why F2 worms are unable to reset small RNAs in response to stress. However, it could be related to the presence of primary small RNAs, which are only produced in response to the dsRNA trigger, and become diluted over generations, and should only be available to the worms during the P0-F1 generations (Almeida et al., 2019). We encountered a similar “critical period” in a previous paper from our lab (Houri-Ze’evi et al., 2016), where we found that external activation of the RNAi machinery can extend the inheritance of an ancestral RNAi response, but only of the external activation occurs within one generation of the RNAi trigger. Similarly, in a recently-published paper from our lab, we found that worms can occupy different epigenetic “states” only at the early generations, and that this state, once established, determines their capability to inherit the small RNA responses across the lineage (Houri-Zeevi et al., 2020). The epigenetic state of the worms could also explain the ability of worms to reset an RNAi trigger only within one generation of the RNAi trigger.

Reviewer #2:Transgenerational inheritance of gene expression states has been demonstrated in plants and animals. In *C. elegans*, transgenerational inheritance often lasts 3-5 generations, and increasing evidence suggests that the inherited gene expression states are mediated by RNA interference pathways. How this inheritance is maintained and "reset" after approximately three generations has been an open question in the small RNA field. Understanding how environmental stress can propagate gene expression states across generations has become an important field of study with regard to human disease, particularly for diseases with fetal origins such as metabolic programming disorders. In a previous publication, this lab showed that the termination of transgenerational gene expression states is a regulated process. In this follow-up manuscript, the authors show that environmental stresses- starvation, high osmolarity, and high temperature- can disrupt the inheritance of gene expression through changes in small RNA levels. The authors also demonstrate that the stress-induced disruption of inherited gene expression is dependent upon the MAPK pathway and the MET-2 histone H3K9 methyltransferase.While I think this manuscript describes an important finding with regards to small RNA inheritance, I think the results need to be more conceptually integrated with previous work.For example, numerous studies have examined transgenerational inheritance after heat stress and the role of MET-2 in establishing H3K9me2 marks in the early embryo and in the adult germ line. How does this study complement or disagree with previous findings? The answer to this question might help in developing a more testable model of how MAPK and MET-2 function are mechanistically linked to small RNA production.

We thank the reviewer for raising this point. In the revised manuscript we delved further into the potential links of our results with the existing literature about MET-2 and H3K9me2 in the early embryo (see details below).

Related to this comment, the authors do not address how this work relates to their previous paper (Houri-Ze'evi et al., Cell), which showed that dsRNA-induced RNAi triggered the RNAi-mediated downregulated of RNAi genes. Does stress in L1 progeny of adults treated with dsRNA further down-regulate these RNAi genes, allowing the "reset" of gene expression to happen earlier?

We thank the reviewer for raising this interesting comment.

To address it, we analyzed mRNA sequencing data from worms that underwent either heat shock, hyperosmotic stress, or starvation. Interestingly, we found that a considerable group of small RNA factors, epigenetic factors, and p-granule proteins, are significantly repressed in response to all stress conditions. In particular, we found that the argonaute genes *hrde-1, rde-1, ergo-1, nrde-3, wago-1,* and *alg-2*, the mutator genes *mut-2* and *mut-16*, and the small RNA factors *rde-4, rrf-3, hrde-4*, and *rde-8*, were all significantly downregulated following exposure to stress. Some of those repressed factors, such as *rde-*1, *rde-*4, *rrf-*3, and *wago-1*, are epigenetic genes which we previously showed were differentially regulated by exposure to an RNAi trigger. We elaborate on these results further in the revised manuscript.

Moreover, we further discuss the connection between this manuscript and our previous paper (Houri-Ze’evi et al., 2016), as well as more recent findings from our lab (Houri-Zeevi et al., 2020; Posner et al., 2019) in our revised manuscript.

In addition, I found the term "reset" to be somewhat of a misnomer, which led to some confusion while reading the manuscript. Dr. Rechavi demonstrated as a postdoc that L1 starvation leads to transgenerational inheritance of a small RNA response to stress. Thus, it seems that stress does not exactly "reset" the small RNA populations to a "normal" state, but rather the worms re-prioritize their small RNA repertoire for a greater need, i.e. starvation/osmolarity/temperature stress trumps silencing a single transgene.

This is a good point.

The main method we used in our paper to observe and quantify the effects of stress on heritable small RNAs was monitoring of ancestral silencing responses that affect fluorescent transgenes, and therefore we refer to these effects throughout the paper as “resetting”. However, the reviewer has a good point and therefore to make the title more precise and to avoid confusion, we changed the title of the manuscript to “Stress Resets Ancestral Heritable Small RNA Responses”.

It is indeed a possibility that worms do not exactly “reset” their small RNA populations but readjust them. Previous studies have shown that the different small RNA populations in the worm are competing over the same silencing and bio-synthesis machinery (Fischer et al., 2011; Gent et al., 2010; Lee et al., 2006). Therefore, it is plausible that synthesis of new small RNAs in response to a stimulus such as stress could come at the expense of other small RNAs, leading to a visible “resetting” of the other small RNA populations.

A similar hypothesis was raised by reviewer #3, and we were glad to investigate it.

We performed an experiment which supports the reviewer’s hypothesis and our 2014 paper. In this experiment, worms were starved either one generation before RNAi treatment, or one generation after RNAi treatment. We observed that starvation blunted the heritable GFP knockdown in both the F1 worms and their progeny, whether the starvation happened before the RNAi trigger or after it.

This result suggests that the resetting we observed may be a result of competition between different populations of small RNAs, and not necessarily the direct removal of small RNAs.

We elaborate on this experiment in Figure 3 and in the text.

The final major comment about the manuscript is that the authors have not fully explored the potential contribution of the insulin signaling pathway to this process, despite reporting an effect on small RNA "resetting" in the daf-2 mutant and a significant overlap of insulin regulated genes with their identified 281 stress-affected genes. Insulin signaling regulates SKN-1 function, which could potentially explain how the authors observe a generalized small RNA response to different stresses (this review might be helpful: doi: 10.1016/j.freeradbiomed.2015.06.008). The authors should perform their transgene expression assay in daf-2;daf-16 and daf-2;skn-1 double mutants to examine the potential role of insulin signaling and DAF-16 in this process. It would also be useful to examine the overlap of their 281 stress-responsive genes with DAF-16 class I and class II genes.

We thank the reviewer for these suggestions. We were glad to address all of them:

1) We performed the transgene expression experiment in *daf-2;daf-16* double mutants. We observed that those mutants reset siRNAs normally following stress. Their response to RNAi appears to be weaker (RNAi resistant). See Figure 5—figure supplement 2.

2) We performed the transgene expression experiment in *daf-2;skn-1* double mutants. We observed that those mutants behaved very similarly to *skn-1* single mutants – they were unable to reset siRNAs in response to stress, and showed a significantly stronger response to RNAi (Eri). See Figure 5—figure supplement 2.

3) When examining the siRNA levels of the 281 stress-responsive genes, we observed no significant overlap or lack thereof between them and the DAF-16 class I or class II genes (Tepper et al., 2013). See the summarized result in Author response table 1:

**Table resptable1:** 

Name	observed	expected	log2 fold enrichment	pvalue	padj	significant?
daf-16 class I genes	1	2.394688	-1.25398	0.305469	0.305469	no
daf-16 class II genes	0	2.493987	-8.48093	0.082492	0.164984	no

Reviewer #3:In the paper entitled "Stress Resets Transgenerational Small RNA Inheritance" Houri-Ze'evi L, Teichman G et al. examine the interaction between multiple heritable phenotypes by knocking down a heritable GFP reporter and examining its interaction with other stresses, such as starvation and high temperature, which cause transgenerationally heritable phenotypes. They demonstrate that exposing worms to stresses inhibits the transgenerational silencing of the GFP reporter strain they use. They further demonstrate that deletion of genes involved in the MAPK pathway, the skn-1 transcription factor and the putative H3K9 methyltranferase met-2 eliminate the differential response in the F1 and F2 generations after exposure to stress and the GFP reporter silencing. They also sequence the small RNAs in the P0 and F1 generation with and without the added stresses. This is an extremely exciting finding but I feel that it should be a little more flushed out before publication. How are these different pathways connected? The system itself should also be a little more rigorously probed. Some detailed suggestions to help ameliorate the paper are listed below.1) For Figure 1 How long does the phenotype persist (what happens in F4 and F5)?

This is a good question. Following this question, we performed an experiment in which we examined the worms for 6 consecutive generations following stress and updated the paper in accordance. We found that the resetting effect lasted as long as the RNAi response was still heritable (5 generations in our case). since some inheritance of the small RNAs is lost in every generation, after 3-5 generations the silencing is completely lost, and the control animals are indistinguishable from animals that were never treated with RNAi. Therefore, after 6 generations there is no observable distance between the stressed and unstressed worms, because both have completely lost the inheritance of the anti-*gfp* siRNAs.

We elaborate on those results further in the revised manuscript (see Figure 1).

If worms are persistently stressed (across many generations in advance of the experiment) does this blunt the effect of the GFP knockdown in the first generation or its transmission? Basically is the reverse true as well?

We thank the reviewer for this interesting suggestion. A similar hypothesis was raised by reviewer #2, and we were glad to investigate it.

We performed an experiment in which worms were starved either three generations before RNAi treatment, one generation before RNAi treatment, or one generation after RNAi treatment. We observed that starvation blunted the heritable GFP knockdown in all F1 worms that were exposed to starvation and their progeny, whether the starvation happened before the RNAi trigger or after it. We did not observe a cumulative effect when worms were starved for multiple generations before RNAi.

This result suggests that the resetting we observed may be a result of competition between different populations of small RNAs, and not necessarily the direct removal of small RNAs.

We elaborate on this experiment in Figure 3 and discuss it further in the text.

2) For Figure 3 it suggests to me that there is only so much bandwidth that the heritable silencing can occupy so if it is activated to respond to a dramatic stress than it isn't going to be as efficient at having heritable silencing of some GFP transgene. It seems like testing the magnitude of the different stresses would be good to help tease this apart. Is there a certain amount of heat shock or starvation which doesn't elicit a heritable response? Can you tune that down to such an extent that there is no effect on the heritable GFP knock-down?

We thank the reviewer for this interesting suggestion.

For each stress condition, we tested multiple durations and magnitudes. Our general finding was that increasing or decreasing the magnitude of the stress conditions does not increase or decrease their effect on the heritable GFP knockdown, but when stress conditions went below a certain magnitude the effect became less consistent (meaning that in some replicates worms would de-silence the GFP transgene in response to the stress, and in other replicates they would not respond at all). We elaborated further on our methodology of choosing stress conditions in the Materials and methods section, and show the stress conditions that lead to resetting in Figure 1—figure supplement 1.

Also I couldn't really figure it out from the text if this was done with one GFP reporter strain or multiple ones? If it was done with multiple strains maybe explain that a little more thoroughly, if it was only done with one strain it would strengthen the paper to do it with another one.

We used a total of 4 fluorescent reporters in the paper: two reporters of RNAi-induced silencing (strains SX1263 and EG6089), a reporter of endo-siRNA-silencing (GR1720), and a reporter of piRNA-silencing (JA1527). We now state it more clearly in the Materials and methods and Results sections.

In particular, the experiment in which we tested *daf-2* mutants was performed using the EG6089 reporter strain instead of the SX1263 transgene that was used in other assays.

3) For Figure 4 it would be nice to delve a little deeper. Does something happen to the MAPK protein levels in response to stress in one generation? It seems as though there isn't much connection between the different mediators of the stress response. Can the authors connect them more either experimentally or at least by describing in more detail their theories.

We thank the reviewer for this question.

To answer this question, we analyzed mRNA sequencing data from worms that were exposed to either heat shock, hyperosmotic stress, or starvation (Dodd et al., 2018; Finger et al., 2019; Schreiner et al., 2019). We did not observe any significant changes in MAPK transcription levels in response to stress. This is to be expected since the MAPK pathway was shown to activate in response to stress through a phosphorylation cascade (Andrusiak and Jin, 2016; Manning et al., 2002). This is also conserved across multiple species (Manning et al., 2002). We elaborate further on this analysis in the text.

To better connect the different mediators of stress-induced resetting of small RNAs, we took multiple approaches:

1) The p38 MAPK pathway was previously shown to regulate SKN-1 through MAPK-dependent phosphorylation, which leads to nuclear localization of the transcription factor SKN-1 (Inoue et al., 2005a). We elaborate on this in the revised manuscript.

2) To discover the downstream effectors of SKN-1 and the MAPK pathway in the stress-induced resetting response, we analyzed mRNA sequencing data from wild-type and skn-1 knockdown worms that were exposed to either control conditions or hyperosmotic stress. We found that exposure to stress represses the expression of a considerable group of epigenetic and small RNA factors. We elaborate on those results further in the revised manuscript.

3) Furthermore, we found that while *skn-1* knockdown on its own did not affect the expression of any notable epigenetic genes, the stress-induced repression of multiple small RNA factors depends upon *skn-1*. We identified a group of small RNA factors that are repressed in response to stress in a *skn-1* dependent manner, which includes, in particular, the argonaute *nrde-3* and the epigenetic factors *rrf-3*, *rde-8* and *nyn-2*. We elaborate on those results further in the revised manuscript.

To identify other potential downstream effectors of *skn-1* that function in resetting, we examine the putative promoter regions of known small RNA factors for the established binding motif sequence of SKN-1. We found that the promoters of multiple small RNA factors contain SKN-1 binding sites, including the RNA-dependent RNA polymerase *rrf-3* and the putative H3K9 histone methyltransferase MET-2. We elaborate on those results further in the revised manuscript.

4) To delve deeper into the mediators of stress response, and to connect the finding in this manuscript to other recent findings from our lab, we examined an additional mutant strain, *hsf-1* mutants. We recently showed that HSF-1 determines the “inheritance state” of worms when initiating a heritable RNAi response (Houri-Zeevi et al., 2020). This “inheritance state” determines the fate of the heritable response (its persistence across generations). Our new experiment showed that *hsf-1* mutants are capable of resetting RNAi inheritance in response to stress. This result stress indicates that HSF-1’s role is likely in the initiation, and not maintenance, of heritable RNAi responses.

4) Again in Figure 5 when discussing different stressors affecting small RNA pools in different ways, it's unclear as to whether this is due to a different magnitude of stress or different pathways. Can the authors delve deeper into this? What is different about the small RNAs? Maybe more description of what the small RNAs are rather than the numbers and types GO analysis of enrichment in each category and the statistics and figure should represent that info.

This is a big question, which is very difficult to address experimentally. Any major stress condition, other than affecting small RNA factors and pathways, affects also many different pathways, and it is difficult to disentangle the different effects. For example, it was shown that even 15-minute starvation of L1 larvae, leads to dramatic changes in expression of 27% in expression (which is more than the number of genes that change expression across the entire larval development) (Maxwell et al., 2012). Additional studies to clearly understand the difference between the stress conditions will be required in the future, but this is outside the scope of this particular work.

[Editors’ note: what follows is the authors’ response to the second round of review.]

Reviewer #1 (Recommendations for the authors):It was unclear whether the germline rescue of sek-2 was tested for its contribution to the observed effects.

Thank you for pointing this out. Germline rescue was not tested in this paper:

1) Since transgene silencing in the germline is also mediated by small RNAs, introducing a transgene rescue might affect the normal function of the RNAi system and will not allow a clear conclusion regarding the role of the rescued gene in stress-induced resetting.

2) The endogenous SEK-1 does not seem to be expressed in the germline and artificially expressing it in the germline would probably generate ectopic germline expression rather than “rescue” of germline expression.

We further clarified in the paper about the tissue-specific rescues that were tested.

Some but not all of the RNA-seq data are described in terms of enrichment and significance; it would be preferable to see this in all the discussions of transcriptional data.

We added enrichment and significance scores to all discussions of transcriptional data.

Reviewer #2 (Recommendations for the authors):1) The figures show that stress treatment of F1 progeny of P0 adults treated with dsRNA resulted in de-silencing of the GFP transgene, presumably through changes in exo-siRNA levels. In Figure 1C, the transgene de-silencing appears complete by the F4 generation, compared to control progeny that do not completely de-silence the transgene until F6 generation. [It would be helpful in these experiments, where appropriate, to perform an ANOVA to compare across generations.] In contrast, when endo-siRNA responses were examined using an endo-siRNA and piRNA transgene sensor (Figure 2), the de-silencing was only observed F1 generation that experienced the stress. These differences indicate two different mechanisms regulating siRNAs during and after stress. Exo-siRNAs appear to be “turned over” much faster due to stress conditions, whereas siRNAs were only affected during the stressed generation. These two mechanisms are lumped together in the manuscript as one mechanism, although the authors do somewhat make distinctions in parts of the Results. This creates problems later in Figure 5D, when rescue of sek-1 is examined using a piRNA sensor strain, while Figure 5C examines the role of mutants regulating exo-siRNAs. For consistency, the experiment in Figure 5D should use a transgene reporter regulated by exo-siRNAs as well. I also think it would be more helpful to the reader to discuss these mechanisms separately in the manuscript.

We appreciate these comments.

ANOVA:

It could have been interesting to perform ANOVA analysis on out data. However, after looking into the matter, we came to the conclusion that performing ANOVA on our fluorescence data would be statistically inappropriate and could yield meaningless or even misleading results. The fluorescence data fails to meet the basic requirements for a two/three-way ANOVA analysis (homoscedasticity and normal distribution of residuals).

Unfortunately, as far as we are aware, there is no appropriate a-parametric alternative to a two-way ANOVA that does not require data homoscedasticity and normal distribution of residuals.

Exogenous RNAi vs. endogenous silencing:

Indeed, we find that exogenous RNAi responses show permanent resetting dynamics compared to endogenous responses which seem to “reestablish” the silencing response in the next generations after stress (most likely due to their ability to be transcribed off the genome). We tried to separate the two phenomena in a clear way in the original version of the manuscript but clarified it further now. We also describe the rescue experiments in a clearer manner. As mentioned in the manuscript, we chose to use the endogenous silencing assay for these rescue experiments since the only established transgene for exogenous silencing (that is not naturally silenced in the germline) overlaps with the transgene expression of the rescue lines (i.e., GFP expression). However, we made sure that the rescue experiments were performed side by side with mutant lines to validate the importance of the examined mutations – and rescues – in endogenous small RNA silencing as well.

2) In Figure 4, the small RNA levels were examined in stressed adults and their non-stressed progeny. They show an overall decrease in sRNAs for all three stresses of stressed adults, but show variable changes for each stress in the non-stressed progeny. This data is largely consistent with the results of Figure 2. Since Figure 3 explored the dynamics of how sRNA populations could be altered by different types of stress, I think it would be informative in Figure 4 to also show the levels of exo-RNAs targeting GFP in these populations. The dynamics of how exo-RNAs and endo-siRNAs may compete for resources during a stress response could be explored more here. In addition, the authors mention the lack of reproducibility among the replicates in Figure 4C. The control lanes look very similar, yet replicates of same stress do not cluster together. Are the replicates for each stress collected at the same time clustering together instead? Perhaps another environmental influence was contributing to these sRNA differences.

We added further discussion regarding the dynamics and possible competition between exogenously and endogenously derived small RNAs.

Regarding the additional environmental factor that might contribute to the variability among different stress replicates: to generate the “cleanest” data, each replicate (including the control groups) in these experiments was collected separately in different days and over the course of three months, with no overlap between different replicates. Thus, we conclude that the observed variability that is observed in the stress groups – but is not observed in the control groups – rises due to the nature of recovery from stress and not due to any batch effects.

3) The nomenclature of a subset of the siRNAs identified in Figure 4C as "stress-reset" I find confusing. I think of a "reset" as return to baseline, whereas in this case, the worms are exhibiting a decrease from baseline for those siRNAs that are maintained over generations to protect germ line integrity.

We changed the term to “stress-reduced”.

4) Figure 1 is blurry and pixelated, which may have occurred during the conversion process.

Thank you for pointing this out. We made sure to have a clear version uploaded.

5) In Figure 1C, 2B, (and others)- it would be helpful to have GFP levels of the transgene with empty vector treatment to compare if populations were actually "reset" or if the populations were gradually de-silencing over generations.

Thank you for this comment. We added the basal level of GFP expression with no RNAi treatment to Figure 1—figure supplement 4.

In more detail: we observed that even six generations after the initiation of RNAi, a large portion of the worm population did not entirely re-express the GFP transgene, and some worms even maintained complete silencing of the transgene (**see** Figure 1—figure supplement 4). Interestingly, in the same experiment, we observed that GFP expression in worms which were stressed several generations ago reached a plateau around the F4 generation, and that progeny of unstressed worms reach said plateau at the F6 generation.

These observations comply with the results of our recent paper (Houri-Zeevi et al., 2020), which showed that, following an RNAi treatment, some lineages of worms tend to completely silence the targeted gene for multiple number of generations – while others “lose” the silencing quite rapidly. The results which were now added to Figure 1—figure supplement 4 suggest that stress is more likely to “reset” heritable silencing in worms that were predisposed to inherit the silencing response for shorter duration.

6) In Figure 5, the mutant names do not line up with the boxes.

Fixed.

7) In Figure 5C- mek-1;sek-1 F2 looks like wildtype F6 in Figure 1. Perhaps this strain has a failure to inherit small RNAs in addition to failure to reset.

This strain is indeed RNAi resistant, as described in Figure 5A and in the manuscript. We added further clarification regarding it in the manuscript.

8) Introduction – nuclear small RNAs *promote modification of* chromatin

We changed the text in accordance.

9) Subsection “Stress, and not any change in growth conditions, leads to theresetting of ancestral small RNA silencing”/Figure 3: it would be helpful to indicate "mild stress" to distinguish it from the harsher stresses used in other figures for clarity

We changed the text and figure in accordance.

10) Are RNAi genes targeted by MET-2, resulting in maintained strong silencing of the transgene?

It’s a very plausible explanation which was explored to some degree at (Rechtsteiner et al., 2016) in which they show H3K9me2 peaks at the promoter region of the Heritable RNAi Deficient-1 (*hrde-1)* gene. However, the general role of MET-2 in silencing and inheritance was not our focus in this study and we hope that future work will explore this possibility further.

11) Results final paragraph – : should be changed to period to start new sentence

Fixed.

12) In the Discussion, it would be helpful to compare the results presented in this manuscript to some of the studies that have examined transgenerational phenotypes resulting from similar stresses.

We added more discussion about comparisons to previous stress-induced transgenerational inheritance studies.

13) Some references incorrectly have too many authors

Fixed.

Reviewer #3 (Recommendations for the authors):The authors for the most part answered my questions and I think the paper is basically ready for publication. I would make a few minor changes as suggested below. Additionally I still feel that the portion with MET-2 is not sufficiently fleshed out nor linked to the rest of the manuscript. I would suggest removing this final figure as the connection to skn-1 etc is tenuous still.

We appreciate this concern. We removed the met-2 part from the main Results section and instead describe these results in the discussion as an opening for future studies.

– First paragraph in subsection “Stress-induced resetting of target-specific small RNAs” ends prematurely "rapamycin exposure and changes in…"– Third paragraph of the same section miscited as Figure 5E should be 4E.– Something messed up happened in the citations. There is a citation Chen Qi, Yan Menghong, etc. which seems to just include a long list of researchers not a specific paper. Similarly the citation Kishimoto S, Uno M,.… is a list of people not in the paper that is cited.

All fixed.

– If the authors insist on including the MET-2 portion of the manuscript (again which I feel is out of place and doesn't yet belong in this paper), when discussing MET-2 and citing Kerr et al., 2014 and work from your own lab it seems appropriate to also cite work from Susan Gasser, Susan Mango, Yang Shi, and Susan Strome, all of whose work came out either before or at the same time.

We added a fuller reference list to the met-2 results in the discussion part.